# PIPS: Path Integral Stochastic Optimal Control for Path Sampling in Molecular Dynamics

## Abstract

We consider the problem of *Sampling Transition Paths*: Given two metastable conformational states of a molecular system, e.g. a folded and unfolded protein, we aim to sample the most likely transition path between the two states. Sampling such a transition path is computationally expensive due to the existence of high free energy barriers between the two states. To circumvent this, previous work has focused on simplifying the trajectories to occur along specific molecular descriptors called Collective Variables (CVs). However, finding CVs is non trivial and requires chemical intuition. For larger molecules, where intuition is not sufficient, using these CV-based methods biases the transition along possibly irrelevant dimensions. In this work, we propose a method for sampling transition paths that considers the entire geometry of the molecules. We achieve this by relating the problem to recent works on the Schrödinger bridge problem and stochastic optimal control. Using this relation, we construct a *path integral* method that incorporates important characteristics of molecular systems such as second-order dynamics and invariance to rotations and translations. We demonstrate our method on commonly studied protein structures like Alanine Dipeptide, and also consider larger proteins such as Polyproline and Chignolin.

## 1 Introduction

Modeling non-equilibrium systems in natural sciences involves analyzing dynamical behaviour that occur with very low probability known as *rare events*, i.e. particular instances of the dynamical system that are atypical. The kinetics of many important molecular processes, such as phase transitions, protein folding, conformational changes, and chemical reactions, are all dominated by these rare events. One way to sample these rare events is to follow the time evolution of the underlying dynamical system using Molecular Dynamic (MD) simulations until a reasonable number of events have been observed. However, this is highly inefficient computationally due to the large time-scales involved in MD simulations, which are typically related to the presence of high energy or entropy barriers between the metastable states. Thus, the main problem is: *How can we efficiently sample trajectories between metastable states that give rise to these rare but interesting transition events?*

Numerous enhanced sampling methods such as steered MD (Jarzynski, 1997), umbrella sampling (Torrie and Valleau, 1977), constrained MD (Carter et al., 1989), transition path sampling (Dellago and Bolhuis, 2009), and many more, have been developed to deal with the problem of *rare events* in molecular simulation. Most of these methods bias the dynamical system with well-chosen geometric descriptors of the transition (analogous to lower dimensional features), called *collective variables* (CVs), that allow the system to overcome high-energy transition barriers and sample these rare events. The performance of these enhanced sampling techniques is critically dependent on the choice of these CVs. However, choosing appropriate CVs for all but the simplest molecular systems is fraught with difficulty, as it relies on human intuition, insights about the molecular system, and trial and error.

A key alternative to sampling these rare transition paths is to model an alternate dynamical system that allows sampling these rare trajectories in an optimal manner (Ahamed et al., 2006; Jack, 2020; Todorov, 2009) or by learning an optimal RL policy for such a transition system Rose et al. (2021).

In this paper, we consider the problem of sampling *rare* transition paths by developing an alternative dynamical system using *path integral stochastic optimal control* (Kappen, 2005; 2007; Kappen and Ruiz, 2016; Theodorou et al., 2010). Our method models this alternative dynamics of the system

by applying an external control policy to each of the atoms in the molecule. We learn the external control policy such that it minimizes the amount of external work needed to overcome the lowest energy barrier and transition the molecular system from an initial meta-stable state to a final one. The method does not require any knowledge of CVs to sample these rare trajectories. Furthermore, we draw connections between sampling rare transition paths and the Schrödinger bridge problem (Schrödinger, 1931; 1932). Subsequently, we show that stochastic optimal control is well suited to solving these problems by extending the work of Kappen and Ruiz (2016) for molecular systems by incorporating Hamiltonian dynamics and equivariance constraints in our path integral SOC method.

Our main contributions in this paper are:

- We demonstrate the equivalence between the problem of sampling transition paths, the Schrödinger bridge problem, and path integral stochastic optimal control (SOC) (§2).
- We develop PIPS, a path integral SOC method that incorporates second order Hamiltonian dynamics with clear physical interpretations of the system (§3).
- In contrast to earlier work, PIPS does not require any knowledge of CVs, which is important for modeling large and complex molecular transitions for which CVs are unknown (§2-3).
- Due to considering second order Hamiltonian dynamics, PIPS seamlessly integrates with common molecular dynamics frameworks such as OpenMM (Eastman et al., 2017).
- We demonstrate the efficacy of PIPS on conformational transitions in three molecular systems of varying complexity, namely Alanine Dipeptide, Polyproline, and Chignolin (§4).

## 2    PRELIMINARIES AND PROBLEM SETUP

Consider a system evolving over time where $\pi(\boldsymbol{x})$ is the distribution of states $\boldsymbol{x}$ and $\pi_i(\boldsymbol{x}_i|\boldsymbol{x}_{i-1})$ a Markovian transition kernel. The distribution of trajectories generated by such a system is given by:

$$\pi\big(\boldsymbol{x}(\tau)\big) := \pi(\boldsymbol{x}_0) \cdot \prod_{i=1}^{\tau} \pi_i(\boldsymbol{x}_i|\boldsymbol{x}_{i-1}). \tag{1}$$

where $\boldsymbol{x}(\tau)$ defines a trajectory of states of length $\tau$ discretized over time into an ordered sequence of states $\boldsymbol{x}(\tau) = \{\boldsymbol{x}_0, \boldsymbol{x}_1, \cdots, \boldsymbol{x}_\tau\}$.

The problem of sampling transition paths involves sampling trajectories from this distribution, $\pi\big(\boldsymbol{x}(\tau)\big)$, with the boundary condition that the initial state $\boldsymbol{x}_0$ and terminal state $\boldsymbol{x}_\tau$ are drawn from pre-specified marginal distributions $\pi_0$ and $\pi_\tau$, respectively. These marginal distributions describe the stable states of the molecular system located at the local minimas of the free energy surface e.g. these stable states can be reactants and products of chemical reactions, or native and unfolded states of protein. Thus, these marginal distributions defining the stable states can be viewed as Dirac delta distributions. Unfortunately, these stable states are often separated by high free energy barriers making the trajectories, $\boldsymbol{x}(\tau)$, sampled starting from $\boldsymbol{x}_0$ to terminate in the target state $\boldsymbol{x}_\tau$ unlikely.

In this paper, we construct a sampling approach that generates trajectories that are still likely under the distribution $\pi\big(\boldsymbol{x}(\tau)\big)$ while also adhering to the boundary conditions by crossing the high free energy barrier by incorporating relevant inductive biases of the system. Formally, we find an alternate dynamical system $\hat{\pi}\big(\boldsymbol{x}(\tau)\big)$ with marginals $\pi_0$ and $\pi_\tau$ that is as close to $\pi\big(\boldsymbol{x}(\tau)\big)$ as possible, i.e.

$$\hat{\pi}^*\big(\boldsymbol{x}(\tau)\big) := \underset{\hat{\pi}(\boldsymbol{x}(\tau)) \in \mathcal{D}(\pi_0, \pi_\tau)}{\arg\min} \mathbb{D}_{\mathsf{KL}}\Big(\hat{\pi}\big(\boldsymbol{x}(\tau)\big) \| \pi\big(\boldsymbol{x}(\tau)\big)\Big) \tag{2}$$

where $\mathcal{D}(\pi_0, \pi_\tau)$ is the space of path measures with marginals $\pi_0$ and $\pi_\tau$. This problem of learning an alternative dynamical system is also known as the Schrödinger Bridge Problem (SBP) (Schrödinger, 1931; 1932). We, thus, take inspiration from recent computational advances for solving SBP (Vargas et al., 2021a; De Bortoli et al., 2021) to develop our solution in §3 to solve the problem of sampling transition paths that can *efficiently* cross the high free energy barriers. Additionally, in this work, we propose an alternative approach to solving SBP using path integral stochastic optimal control that lends itself well to modelling the chemical nature of our problem.

In the next section, we will set the stage for this novel approach by first relating the problem of sampling transition paths as a path integral stochastic optimal control problem. Subsequently, we will

establish an equivalence between learning an alternative dynamical system for sampling transition paths, Schrödinger bridge problem, and stochastic optimal control.

## 2.1 SAMPLING TRANSITION PATHS THROUGH STOCHASTIC OPTIMAL CONTROL

The original dynamics of the system, as given in eq. (1), can be reformulated as a stochastic process:

$$\mathrm{d}\boldsymbol{x}_t = \boldsymbol{f}(\boldsymbol{x}_t, t)\,\mathrm{d}t + \boldsymbol{G}(\boldsymbol{x}_t, t) \cdot \mathrm{d}\boldsymbol{\varepsilon}_t, \qquad t \in [0, \tau] \tag{3}$$

where $\boldsymbol{f} : \mathbb{R}^d \times \mathbb{R}^+ \to \mathbb{R}^d$ and $\boldsymbol{G} : \mathbb{R}^d \times \mathbb{R}^+ \to \mathbb{R}^{d \times d}$ are deterministic functions representing the drift and volatility of the system. The stochastic process $\boldsymbol{\varepsilon}_t$ is a Brownian motion with variance $\nu$.

As we stated before, the system dynamics in Equation (3) is insufficient for sampling molecular transition paths as they do not adhere to the boundary conditions imposed by the problem. We, thus, add an external bias potential (or control) $\boldsymbol{u}(\boldsymbol{x}_t, t) \in \mathbb{R}^d \times \mathbb{R}^+$ to the system that pushes the molecule over the transition state barriers. We can write the dynamics of this new system as follows:

$$\mathrm{d}\boldsymbol{x}_t = \boldsymbol{f}(\boldsymbol{x}_t, t)\,\mathrm{d}t + \boldsymbol{G}(\boldsymbol{x}_t, t) \cdot \Big(\boldsymbol{u}(\boldsymbol{x}_t, t)\,\mathrm{d}t + \mathrm{d}\boldsymbol{\varepsilon}_t\Big), \qquad t \in [0, \tau] \tag{4}$$

Given a trajectory $\boldsymbol{x}(\tau) = (\boldsymbol{x}_0, \cdots, \boldsymbol{x}_\tau) \in \mathbb{R}^{\tau \times d}$ generated through the SDE in eq. (4), we define the cost of this trajectory under control $\boldsymbol{u}$ following Kappen (2007); Theodorou et al. (2010) as:

$$C(\boldsymbol{x}(\tau), \boldsymbol{u}, \boldsymbol{\varepsilon_t}) = \frac{1}{\lambda}\Big(\varphi(\boldsymbol{x}_\tau) + \sum_{t=0}^{\tau} \frac{1}{2}\boldsymbol{u}(\boldsymbol{x}_t, t)^T \boldsymbol{R}\boldsymbol{u}(\boldsymbol{x}_t, t) + \boldsymbol{u}(\boldsymbol{x}_t, t)^T \boldsymbol{R}\boldsymbol{\varepsilon}_t\Big) \tag{5}$$

where $\varphi$ denotes the terminal cost, $\lambda$ is a constant and $\boldsymbol{R}$ is the cost of taking action $\boldsymbol{u}$ in the current state and is given as a weight matrix for a quadratic control cost. The goal then becomes to find the optimal control $\boldsymbol{u}^*$ that minimizes the expected cost in Equation (5):

$$\boldsymbol{u}^* = \arg\min_{\boldsymbol{u}} \mathbb{E}_{\tau, \boldsymbol{\varepsilon}_t}\big[C(\boldsymbol{x}(\tau), \boldsymbol{u}, \boldsymbol{\varepsilon_t})\big] \tag{6}$$

where the expectation is taken over trajectories $\tau$ sampled using the SDE under control $\boldsymbol{u}$. Before proceeding further, a couple of remarks are in order:

**Remark 1.** *We note that the control $\boldsymbol{u}(\boldsymbol{x}_t, t)$ in Equation (4) does not operate directly on the system dynamics but is controlled through the same control matrix $\boldsymbol{G}$ as the Brownian motion. This formulation is highly crucial for our method, PIPS, to incorporate system specific second order Hamiltonian dynamics as we will show in Section 3.*

**Remark 2.** *The last term in the cost function in eq. (5) relating the Brownian motion and the control is unusual and devoid of a clear intuition. However, this term plays an important role when relating the cost to a KL-divergence which we will establish next. Additionally, as discussed in Thijssen and Kappen (2015), the additional cost vanishes under expectation ($\mathbb{E}_{\tau, \boldsymbol{\varepsilon}_t}[u(\boldsymbol{x}_t, t)^T \boldsymbol{R}\boldsymbol{\varepsilon}_t] = 0$) and thus, does not influence the optimal control $\boldsymbol{u}^*$ given by eq. (6)*

**Relation to sampling transition paths:** Interestingly, the objective in Equation (6) is exactly related to the problem of sampling transition paths as given in Equation (2). As Kappen and Ruiz (2016) establish, Equation (4) defines a probability distribution $\pi_{\boldsymbol{u}}\big(\boldsymbol{x}(\tau)\big)$ over trajectories $\boldsymbol{x}(\tau)$ through:

$$\pi_{\boldsymbol{u}}\big(\boldsymbol{x}(\tau)\big) = \prod_{t=0}^{\tau} \mathcal{N}(\boldsymbol{x}_{t+1}|\boldsymbol{\mu}_t, \Sigma_t) \tag{7}$$

with $\boldsymbol{\mu}_s = \boldsymbol{x}_s + \boldsymbol{f}(\boldsymbol{x}_s, s)\,\mathrm{d}t + \boldsymbol{G}(\boldsymbol{x}_s, s)(\boldsymbol{u}(\boldsymbol{x}_s, s)\,\mathrm{d}t)$ and $\Sigma_s = \boldsymbol{G}(\boldsymbol{x}_s, s)^T \nu \boldsymbol{G}(\boldsymbol{x}_s, s)$.

For different $\boldsymbol{u}$, these distributions are related through the Girsanov Theorem (Cameron and Martin, 1944). As shown in appendix B, if we make the common assumption that the control cost $\boldsymbol{R}$ and the variance of the Brownian motion $\nu$ are inversely correlated as $\lambda \boldsymbol{R}^{-1} = \nu$, we can obtain:

$$\log \frac{\pi_{\boldsymbol{u}}\big(\boldsymbol{x}(\tau)\big)}{\pi_0\big(\boldsymbol{x}(\tau)\big)} = \frac{1}{\lambda} \sum_{t=0}^{\tau} \frac{1}{2}\boldsymbol{u}(\boldsymbol{x}_t, t)^T \boldsymbol{R}\boldsymbol{u}(\boldsymbol{x}_t, t) + \boldsymbol{u}(\boldsymbol{x}_t, t)^T \boldsymbol{R}\boldsymbol{\varepsilon}_t \tag{8}$$

where $\pi_0\big(\boldsymbol{x}(\tau)\big)$ denotes the distribution over trajectories with no control i.e. $\boldsymbol{u} = 0$ (§eq. (1)). This assumption of relating the control cost and the variance of the Brownian motion is a common trait of control problems referred to as *Path Integral Stochastic Optimal Control* (Kappen, 2005).

We observe that the right-hand side of eq. (8) can also be found in the definition of the control cost in eq. (5), including the additional cost term related to the Brownian noise. As Kappen and Ruiz (2016) show, we can thus use eq. (8) to rewrite the objective in Equation (6) as:

$$\pi_{\boldsymbol{u}^*} = \arg\min_{\pi_{\boldsymbol{u}}} \mathbb{E}_{\boldsymbol{x}(\tau)\sim\pi_{\boldsymbol{u}}}\Big[\frac{1}{\lambda}\varphi(\boldsymbol{x}_\tau)\Big] + \mathbb{D}_{\mathsf{KL}}\Big(\pi_{\boldsymbol{u}}\big(\boldsymbol{x}(\tau)\big)\|\pi_0\big(\boldsymbol{x}(\tau)\big)\Big) \tag{9}$$

This objective is an approximation of the Schrodinger Bridge formulation in Equation (2) where the constraints on the marginal distributions are replaced by a regularization term in the form of the terminal cost. Therefore, when the terminal cost dominates the KL-divergence term above, it enforces the target boundary constraints of the problem. Before we discuss an algorithm to learn this optimal policy in Equation (9) next, we end this part with a remark:

**Remark 3.** *This connection between the Schrödinger Bridge Problem and stochastic optimal control has been previously established (Chen et al., 2016; Pavon et al., 2021). However, through the formulations in Equations (2) and (9), we also establish the equivalence between sampling transition paths, Schrödinger bridge problem, and stochastic optimal control. This allows us to utilize solutions for finding the optimal control in Equation (9) for the aforementioned problems.*

**Optimal Control Policy:** Kappen and Ruiz (2016) introduced the Path Integral Cross Entropy (PICE) method for solving Equation (9). The PICE method derives an explicit expression for the optimal policy and distribution $\pi_{\boldsymbol{u}^*}$ when $\lambda = \nu\boldsymbol{R}$ given by:

$$\pi_{\boldsymbol{u}^*} = \frac{1}{\eta(\boldsymbol{x}, t)}\pi_{\boldsymbol{u}}\big(\boldsymbol{x}(\tau)\big)\exp(-C(\boldsymbol{x}(\tau), \boldsymbol{u})) \tag{10}$$

where $\eta(\boldsymbol{x}, t) = \mathbb{E}_{\boldsymbol{x}(\tau)\sim\pi_0}[\exp(-\frac{1}{\lambda}\varphi(\boldsymbol{x}_\tau)]$ is the normalization constant. This establishes the optimal distribution $\pi_{\boldsymbol{u}^*}$ as a reweighing of any distribution induced by an arbitrary control $\boldsymbol{u}$. Similar to importance sampling, depending on the choice of the proposal distribution $\pi_{\boldsymbol{u}}$, the estimator variance can greatly differ. Thus, the objective is to find the $\boldsymbol{u}$ that best approximates $\boldsymbol{u}^*$.

PICE, subsequently, achieves this by minimizing the KL-divergence between the optimal controlled distribution $\pi_{\boldsymbol{u}^*}$ and a parameterized distribution $\pi_{\boldsymbol{u}_\theta}$ using gradient descent as follows:

$$\frac{\partial\mathbb{D}_{\mathsf{KL}}(\pi_{\boldsymbol{u}^*}|\pi_{\boldsymbol{u}_\theta})}{\partial\theta} = -\frac{1}{\eta}\mathbb{E}_{\boldsymbol{x}(\tau)\sim\pi_{\boldsymbol{u}_\theta}}[\exp(-C(\boldsymbol{x}(\tau), \boldsymbol{u}_\theta))\sum_{t=0}^{\tau}(\boldsymbol{R}\varepsilon_t \cdot \frac{\partial\boldsymbol{u}_\theta}{\partial\theta})] \tag{11}$$

Similar to the optimal control in eq. (10), the gradient used to minimize the KL-divergence is found by reweighing for each sampled trajectory, $\boldsymbol{x}(\tau)$, the gradient of the control policy $\boldsymbol{u}_\theta$ by the cost of said trajectory. Algorithm 1 in the appendix provides a method for finding this gradient and training the policy $\boldsymbol{u}_\theta$. Thus, PICE provides an iterative gradient descent method to learn a parameterized policy $\boldsymbol{u}_\theta$ and subsequently a distribution over paths $\boldsymbol{x}(\tau)$. We can then use this learned control, $\boldsymbol{u}_\theta$, to approximate the solution for sampling transition paths as well as the Schrödinger bridge problem.

In this section, we set up our main problem of sampling transition paths and established its relationship to both the Schrödinger bridge problem and stochastic optimal control. Subsequently, we discussed an iterative gradient descent based method for solving the optimal control problem. In the next section, we will extend this iterative algorithm to consider the entire geometry of the molecular system by incorporating Hamiltonian dynamics using an augmented state space $\boldsymbol{x}_t$, and symmetries by learning a policy network $\boldsymbol{u}_\theta$.

## 3 PATH INTEGRAL OPTIMAL CONTROL FOR SAMPLING TRANSITION PATHS

We consider a molecule consisting of $n$ atoms with an initial and final configuration $\boldsymbol{r}_0 \in \mathbb{R}^{3\times n}$ and $\boldsymbol{r}_\tau \in \mathbb{R}^{3\times n}$ i.e. we are given a vector defining the 3D positions of each atom in the molecule. Thus, a direct method to sample transition paths $\boldsymbol{r}(\tau)$ for this problem is to learn a control $\boldsymbol{u}_\theta$ acting directly on the positions $\boldsymbol{r}$ of the molecule using the iterative gradient descent method discussed in Section 2.

However, the collective behaviour of the atoms and molecules are governed by classical molecular dynamics i.e. Newtonian equations of motion:

$$\mathrm{d}\boldsymbol{r} = \boldsymbol{v}(t)\,\mathrm{d}t, \quad \text{and,} \quad \mathrm{d}\boldsymbol{v} = \boldsymbol{a}(t)\,\mathrm{d}t \tag{12}$$

where $\boldsymbol{v}(t) \in \mathbb{R}^{3 \times n}$ is the velocity and $\boldsymbol{a}(t) \in \mathbb{R}^{3 \times n}$ is acceleration given by $\boldsymbol{a}(t) = \nabla_r \boldsymbol{U}(\boldsymbol{r})/\boldsymbol{m}$ where $\boldsymbol{U}(\boldsymbol{r})$ is the potential energy of the system and $\boldsymbol{m}$ is the mass. The potential energy of a system is defined by a parameterized sum of pairwise empirical potential functions, such as harmonic bonds, angle potentials, inter-molecular electrostatic and Van der Waals potentials. In our work, we compute this potential energy using the OpenMM framework (Eastman et al., 2017). Therefore, in light of Equation (12), we need to adapt the dynamical system defined in Equation (4) to incorporate these molecular dynamics.

**Incorporating second order dynamics:** Formally, we incorporate the second order dynamics of the system defined above by considering an augmented state space: Let $\boldsymbol{x}_0 := (\boldsymbol{r}_0, \boldsymbol{v}_0) \in \mathbb{R}^{3 \times n} \times \mathbb{R}^{3 \times n}$ be the initial configuration of the system defining the initial positions and velocities of each atom and $\boldsymbol{x}_\tau := (\boldsymbol{r}_\tau, \boldsymbol{v}_\tau)$ be the final configuration. We, thus, model the dynamical system in Equation (4) as:

$$\underbrace{\begin{pmatrix} \mathrm{d}\boldsymbol{r}_t \\ \mathrm{d}\boldsymbol{v}_t \end{pmatrix}}_{\mathrm{d}\boldsymbol{x}_t} = \underbrace{\begin{pmatrix} \boldsymbol{v}_t \\ -\nabla_{\boldsymbol{r}_t}\boldsymbol{U}(\boldsymbol{r}_t) \end{pmatrix}}_{\boldsymbol{f}(\boldsymbol{x}_t,t)}\mathrm{d}t + \underbrace{\begin{pmatrix} \boldsymbol{0}_{3n} \\ \mathbb{I}_{3n} \end{pmatrix}}_{\boldsymbol{G}(\boldsymbol{x}_t,t)} \cdot \Big(\boldsymbol{u}(\boldsymbol{x}_t,t)\,\mathrm{d}t + \mathrm{d}\boldsymbol{\varepsilon}_t\Big), \qquad t \in [0,\tau] \tag{13}$$

Due to the choice of $\boldsymbol{G}(\boldsymbol{x}_t, t)$ in Equation (13) above, the additional bias force, $\boldsymbol{u}(\boldsymbol{x}_t, t)$, applied to the system only influences the acceleration and velocity of the atoms and does not act directly on the positions of the atoms. $\mathrm{d}\boldsymbol{r}_t$ is solely influenced by the velocity $\boldsymbol{v}_t$, thus conforming to the classical molecular dynamics of the system as given in Equation (12).

Unfortunately, this new dynamical system in Equation (13) leads to a singular covariance matrix, $\Sigma_t$ in eq. (7) due to the choice of $\boldsymbol{G}$. However, due to the conditional independence of $\boldsymbol{r}_{t+1}$ given $(\boldsymbol{r}_t, \boldsymbol{v}_t)$, we are able to factorize the distribution in Equation (7) which circumvents the singularity of the covariance matrix. Due to space constraint, we provide details and derivations in Appendix B.1.

**Remark 4.** *We note here that second order dynamics have been considered before for stochastic optimal control by Kappen (2007) for a synthetic spring experiment in one dimension and SBP by Vargas et al. (2021a) for modelling motion. Our formulation of incorporating second-order dynamics here is distinct and more practical than these previous works. Courtsey of eq. (13), we have a clear physical interpretation of the control $\boldsymbol{u}$ as an external physical force by limiting it to act linearly on the velocity $\boldsymbol{v}$. This is interesting for downstream applications of the sampled transition paths such as reconstructing free-energy surfaces. Additionally, it also simplifies incorporating the control with MD simulation software like OpenMM which we will discuss in detail in section 4.*

**Invariance to rotations and translations:** Secondly, the molecules in consideration are invariant w.r.t. translations and 3D rotations i.e. the molecular orientations achieved along a transition path need to incorporate this equivariance w.r.t. the SE(3) group. For this purpose, we need to make the terminal cost function, $\varphi(\boldsymbol{x}_\tau)$, in Equation (5) to be equivariant. We enforce this by defining the terminal cost as the exponentiated pairwise distance between atoms which is commonly used distance metric (Shi et al., 2021) that is invariant to rotations and translations i.e. $\varphi(\boldsymbol{r}_t) = \exp\sum_{i,j}^{n}\big(d_{ij}(\boldsymbol{r}_t) - d_{ij}(\boldsymbol{r}_\tau)\big)^2$ where $d_{ij}(\boldsymbol{r}_t) = \|(\boldsymbol{r}_t)_i - (\boldsymbol{r}_t)_j\|_2^2$.

**Physics inspired policy network ($\boldsymbol{u}_\theta$):** The main learnable component of our PIPS method (as described by eq. (13)) for sampling transition paths is the policy network $\boldsymbol{u}_\theta$. Following the discussion above and formalized in Equation (13), we can interpret the control $\boldsymbol{u}_\theta$ as an additive bias force applied to the system. In this work, we consider two different design approaches to modelling $\boldsymbol{u}_\theta$. In our first approach, we model $\boldsymbol{u}_\theta$ as a neural network that predicts the bias force on the system in which case the velocity evolves as $\mathrm{d}\boldsymbol{v} = (\nabla_{\boldsymbol{r}_t}\boldsymbol{U}(\boldsymbol{r}_t) + \boldsymbol{u}_{\theta,t})\,\mathrm{d}t$. Alternatively, in our second approach, we model $u_\theta$ as a network predicting the bias potential energy. In this case, the corresponding force, $\boldsymbol{F}(\boldsymbol{r}_t)$, applied to the system is calculated by backpropagating through the network, $\boldsymbol{F}(\boldsymbol{r}_t) := \nabla_{\boldsymbol{r}_t} u_{\theta,t}$. The change in velocity is then given by $\mathrm{d}\boldsymbol{v} = \big(\nabla_{\boldsymbol{r}_t}\boldsymbol{U}(\boldsymbol{r}_t) + \boldsymbol{F}(\boldsymbol{r}_t)\big)\,\mathrm{d}t$. Additionally, $\boldsymbol{u}_\theta$ or $u_\theta$ can be implemented using recent advances in physics inspired equivariant neural networks (Cohen and Welling, 2016; Satorras et al., 2021) that take into account the SE(3) symmetry of the system. We provide details for training the control network $\boldsymbol{u}_\theta$ in Appendix A.

|  | $\tau$ fs | Temp. K | EPD ($\downarrow$) nm $\times 10^{-3}$ | THP ($\uparrow$) % | ETP ($\downarrow$) kJ mol$^{-1}$ |
|---|---|---|---|---|---|
| Force Prediction | 500 | 300 | 2.07 | 41.1 % | 0.68 |
| Energy Prediction | 500 | 300 | 1.25 | 89.2 % | -5.21 |
| MD w. fixed timescale | 500 | 300 | 7.92 | 0% | - |
|  | 500 | 1500 | 7.47 | 0% | - |
|  | 500 | 4500 | 6.33 | 0% | - |
|  | 500 | 9000 | 6.82 | 1.7 % | 1019.83 |
| MD w/ fixed timescale | 34810 | 1500 | 1.88 | 100% | 551.51 |
|  | 48683 | 4500 | 2.01 | 100% | 1647.35 |

Table 1: Benchmark scores for the proposed method and extended MD baselines. From-left-to-right: Time-horizon $\tau$ representing the trajectory length (note that we take one policy step every 1 fs), simulation temperature, Expected Pairwise distance (EPD), Target Hit Percentage (THP), and Energy Transition Point (ETP). ETP can only be calculate when a trajectory reaches the target. All metrics are averaged over 1000 trajectories except for MD w/ fixed timescale which is ran only for 10 trajectories.

## 4 EXPERIMENTS

We evaluate our path integral stochastic optimal control method for sampling transition paths with three different molecular systems, namely (i) **Alanine Dipeptide**, a small amino acid with well-studied transition paths, (ii) **Polyproline**, a small protein with two distinct conformations with different helix orientations, and (iii) **Chignolin**, an artificial mini-protein studied to understand the folding process of proteins. We begin by detailing the experimental setup below.

**Molecular Dynamics Simulation:** As we discussed in section 3, we use the OpenMM framework to simulate the molecular dynamics following Equation (13). Crucially, by considering the second order dynamics, the control acts linearly on the molecular potential function in this formulation of the molecular dynamics. This allows us to implement the resulting control as a bias potential that is acting on the system in addition to the molecular potential. At every step of the Molecular Dynamics this bias potential is calculated using our PyTorch implementation of the control and then passed to OpenMM as a custom external force. Implementing the control this way thus allows us to use the optimized configuration capabilities of OpenMM, such as forcefield definitions (the potential function description) and integrators (for the time-discretization of our dynamics). We report the molecule specific OpenMM configuration in appendix C. Generally, we run our simulations at 300 K.

**Policy Network, $u(x_t, t)$:** We implement the policy network as a 6 layer MLP with ReLU activation for all our experiments below. The width of the layers of the policy network is dependent on the number of atoms in the molecule under consideration. We implement all code in Pytorch. We ran the experiments on a single GPU (either an NVIDIA RTX3080 or RTX2080). Our code, including a full stand-alone notebook re-implementation, is available here: https://github.com/pips4anonymous/pips-anonymous.

### 4.1 ALANINE DIPEPTIDE

Alanine Dipeptide is an extensively studied molecule (Tobias and Brooks III, 1992; Rossky and Karplus, 1979; Head-Gordon et al., 1991; Swenson et al., 2018) for developing and testing enhanced sampling methods due to ready availability of its two CVs $(\phi, \psi)$. The conformation transition for Alanine Dipeptide can thus be understood in terms of these two dihedral angles $\phi$ and $\psi$ as displayed in Figure 1A. Prior work has, thus, focused on transforming from the initial configuration (see Figure 1A) to the final configuration (Figure 1E) by rotating these CVs. As we discussed previously, a major advantage of our method is that we do not require the knowledge of CVs to sample a transition path. However, in our experiment for Alanine Dipeptide, we will use these CVs to compare the quality of the trajectory sampled by our method.

**Setup:** For our experiment, we consider both the design choices for the policy network, $u_\theta(x_t, t)$, discussed in Section 3 i.e. directly predicting the force and predicting the energy. We trained the policy networks for 15,000 roll-outs with a time horizon of 500 fs each consisting of 16 samples. A gradient update was made to the policy network after each roll-out with a learning rate of $10^{-5}$. The Brownian motion has a standard deviation of 0.1.

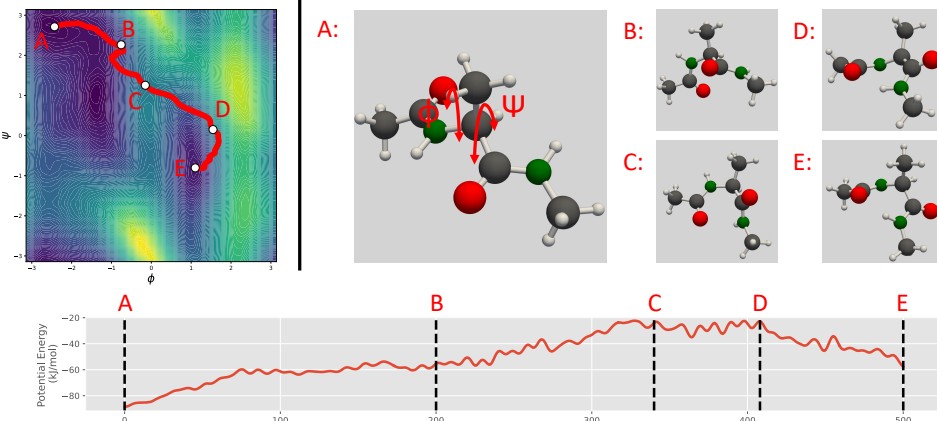

Figure 1: Visualization of a trajectory sampled with the proposed method. *Left:* The sampled trajectory projected on the free energy landscape of Alanine Dipeptide as a function of two CVs *Right:* Conformations along the sampled trajectory: A) starting conformation showing the CV dihedral angles, B-D) intermediate conformations with C being the highest energy point on the trajectory, and E) final conformation, which closely aligns with the target conformation. *Bottom:* Potential energy during transition. Letters represent the same configurations in the transition.

**Baseline and evaluation metrics:** We compare our method to MD simulations with extended time-horizon and increased system temperatures to sample transition paths. To our knowledge, there are no fixed quantitative metrics in the literature to compare different methods that sample transition paths. Thus, we introduce here three metrics to evaluate the quality of transition paths: (i) *Expected Pairwise Distance* (EPD) measures the euclidean distance between the final conformation in the trajectory and the target conformation, reflecting the goal of the transition to end in the target state, (ii) *Target Hit Percentage* (THP) assures that the final configuration is also close in terms of CVs by measuring the percentage of trajectories correctly transforming these CVs, and (iii) *Energy Transition Point* (ETP) which evaluates the capacity of each method to find transition paths that cross the high-energy barrier at a low point by taking the maximum potential energy of the molecule along the trajectory. A good trajectory will be one that passes through the minimal high-energy barrier and ETP aims to measure this. We provide more details in Appendix C.2.1.

**Results:** We first visualize the trajectory generated by the *energy prediction* policy in Figure 1 and defer the visualization for the *force prediction* policy to Appendix C.2.2. The trajectory in Figure 1 demonstrates that the control policy transforms the molecule from the initial position (A) to the final position (E) by transitioning over the barrier with the least energy at (C). Interestingly, the trajectory follows the expected transitions in the CVs without them being explicitly specified e.g. the transition path visualized on the left in Figure 1 shows that the molecule first rotates the dihedral angle associated with CV $\phi$ in (A → B), then gradually rotates along both $\psi$ and $\phi$ in (B → C → D), and finally rotates $\psi$ in (D → E) to reach the final configuration. As expected, we observe that the potential energy goes up during the transition until it reaches the top of the energy barrier (C). After this point, the molecule settles down in its new low-energy state.

Next, in Table 1, we compare the performance of the trajectories sampled using the *force* and *energy* predicting policy networks with MD simulations on the metrics introduced before. We find that the trajectories generated by both the policy networks outperform the MD baselines, but the more physics-aligned energy predicting policy performs best under our metrics. This policy network consistently reaches the target conformation both in terms of full geometry and the CVs orientation. Furthermore, our policy network generates these trajectories in a significantly shorter time than temperature enhanced MD simulations without a fixed timescale. When we do limit MD to run for the same timescale as the proposed method, we found that, in contrast to the proposed method, temperature enhanced MD simulations are unable to generate successful trajectories.

## 4.2 POLYPROLINE HELIX

Polyproline is a helix-shaped protein structure that consists of repeating proline residues. Polyproline helix can form two different conformations namely *Polyproline-I* (PP-I) and *Polyproline-II helix*

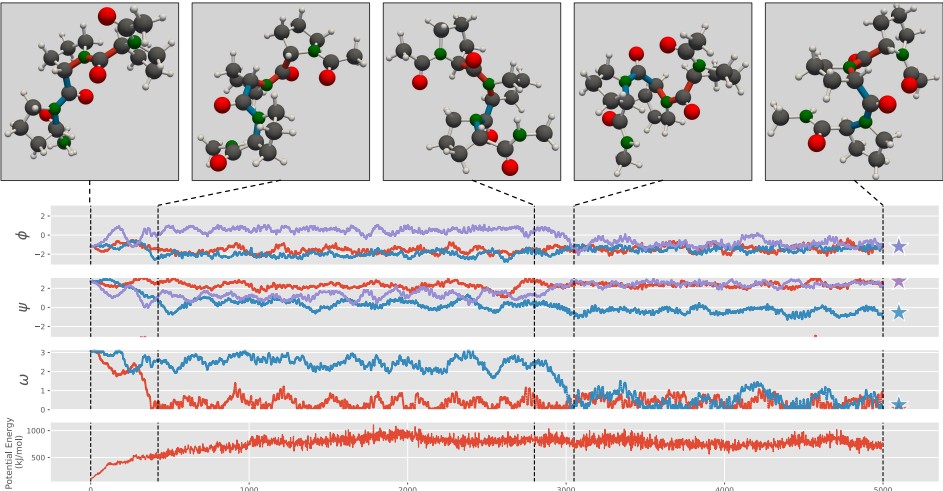

Figure 2: Visualization of the Polyproline transformation from PP-II to PP-I. *From-top-to-bottom* 5 stages of the transition, $\psi$ CVs, $\phi$ CVs, $\omega$ CVs, and Potential Energy. For the CVs multiple instances of the same dihedral angles can be found in a single molecule. Stars indicate target CV states. Colored bonds represent the bonds involved in the $\omega$ CV.

(PP-II) (Moradi et al., 2009; 2010). These conformations can be distinguished by their respective helix rotation. PP-I forms a compact right-handed helix due to its peptide bonds having cis-isomers while PP-II has trans-isomer peptide bonds and forms a left-handed helix. Furthermore, the backbone of the polyproline helix also contains two different dihedral angles. We will refer to these peptide bonds and dihedral-angles as the $\omega$, $\phi$ and $\psi$ CVs respectively. Polyproline can have varying lengths due to its repeated structure. In our experiment, we consider the polyproline trimer with 3 proline residues transitioning from PP-II to PP-I.

**Setup:** The policy network was trained over 500 rollouts with 25 samples each using a learning rate of $3 \times 10^{-5}$ and a standard deviation of 0.1 for the Brownian motion.

**Results:** We visualize the transformation of the three collective variables $(\omega, \phi, \psi)$ as well as the corresponding potential energy of the conformation in Figure 2 for a sampled transition path from our trained policy network. The $\omega$ CV admits the biggest change for the transition from PP-I going from 180° to 0°. We observe that the transition path sampled by our method aligns with the expected changes in CVs in spite of our method not containing any knowledge about these CVs. Figure 2 shows that the peptide bonds transition from a trans-isomer to a cis-isomer state at steps 450 and 3,000. We notice the biggest changes in CVs at these steps in Figure 2. We also note that in addition to the change in the peptide bonds, the final conformation differs from the initial in one of the $\psi$-dihedral angles. Technically, PP-I has $\psi$-dihedral angles similar to PP-II, but as a result of the inherent noise of MD our target conformation was sampled with a slight rotation here as well. Interestingly, our method successfully learned to sample transition paths terminating in a similar perturbed state. This indicates that our proposed method is resilient to target states not having a minimal-energy configuration.

### 4.3 CHIGNOLIN

Chignolin is a small $\beta$-hairpin protein constructed artificially to study protein folding mechanisms (Honda et al., 2004; Seibert et al., 2005). Due to its small size, its folding process is easier to study than larger scale proteins while being similar enough to shed light on this complex process. In contrast to Alanine Dipeptide and Polyproline, there is no agreement on the transition mechanism describing the folding of Chignolin. Both the CVs involved (Satoh et al., 2006; Paissoni and Camilloni, 2021), as well as the sequence of steps (Harada and Kitao, 2011; Satoh et al., 2006; Suenaga et al., 2007; Enemark et al., 2012) describing the folding process have multiple different interpretations. Thus, methods that do not require prior knowledge of CVs are particularly useful to study this protein.

**Setup:** We sample transition paths between the folded and unfolded state of the Chignolin protein using a total time horizon of $5000\,\mathrm{fs}$. Note that the typical folding time of Chignolin is recorded to be

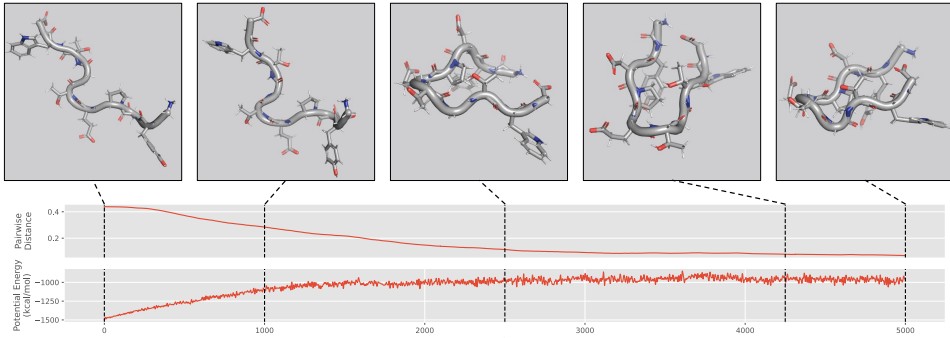

Figure 3: Visualization of the Chignolin folding process. *Top:* 5 stages of the folding process, *Middle:* Pairwise distance wrt to the target conformation of the molecule, *Bottom:* Potential Energy.

$0.6\,\mu s$ (Lindorff-Larsen et al., 2011). The policy network is trained over 500 rollouts of 16 samples with a learning rate of $1 \times 10^{-4}$ and standard deviation of $0.05$ for the Brownian motion.

**Results:** In Figure 3, we visualize the transition of Chignolin at 5 different timesteps during the transition path. We observe that to transition the protein from its low energy unfolded state to the folded conformation, the proposed method guides the protein into a region of higher energy. This increase is initially more steep ($0 \rightarrow 1500$) than in the later stages. Additionally, most of the finer-grained folding ($2500 \rightarrow 4000$) occurs with a high potential energy before settling into the lower-energy folded state. We notice that for the restricted folding time we use in our experiments ($5000\,\mathrm{fs}$ vs $0.6\,\mu s$), the molecule does not end at the final configuration but reaches close to it as shown by the plot on pairwise distance. Furthermore, the learned policy network is able to transition through the high energy transition barrier in this restricted time. We do not encounter this for molecules with a shorter natural transition time (as illustrated by the potential energy of Alanine Dipeptide in fig. 1).

## 5 DISCUSSIONS, LIMITATIONS, AND FUTURE WORK

In this work, we proposed a path integral stochastic optimal control method for the problem of sampling rare transition paths for molecular systems that incorporates the Hamiltonian dynamics and equivariance of the system. In passing, we showed an equivalence between the problem of sampling transition paths, stochastic optimal control, and the Schrödinger bridge problem. We empirically tested our method on three different molecular systems of varying sizes and demonstrated that it was able to sample transition paths on the full geometry of the system without biasing along CVs.

One observed limitation of the proposed method is that for molecules with long natural transition times, we observe the transitions to not converge to the configuration of minimal energy after crossing the high-energy transition barrier. We hypothesize that this is due to the method operating on a reduced time horizon (e.g. $5000\,\mathrm{fs}$ instead of $0.6\,\mu s$ in the case of Chignolin), or due to the terminal control cost function not requiring the velocity to be zero at the end of the transition. Nevertheless, we note that the method is successful in transitioning the molecules over the high energy barriers as exemplified by the known CVs changing appropriately.

There are many exciting directions for future work. Most importantly, we are excited to see how research from other machine learning fields can be used to develop, possibly more efficient, methods for sampling trajectories between molecular conformations. Given the vast literature on Stochastic Optimal Control theory, Schrodinger Bridge samplers, and other topics such as Covariance Control (Yin et al., 2021; Hotz and Skelton, 1987) and Reinforcement Learning (Das et al., 2021) we hope that our path-integral based method can serve as a starting point for machine learning based solutions for the problem introduced in our work. Following this, these approaches and their sampled trajectories, could be used for solving related problems in chemistry. For example, our experiments showed that the molecules transitioned along the CVs correctly in spite of not having any information about the CVs. It will be interesting to see if we can infer these CVs from the learned policy and dynamics of the systems. Lastly, our method can have implications for training diffusion models within a fixed time-scale by additionally learning the control policy to transform one distribution into another. First explorations of this line of thinking are presented in (Vargas et al., 2021b; Zhang and Chen, 2021).

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

# A  ALGORITHMS

## A.1  LEARNING

---

**Algorithm 1:** Training Policy $u_\theta$

---

**Input:** $r_0, r_T$: *Initial and target molecular positions,*
  $U(\cdot)$: *Potential Energy function,*
  $\varphi(\cdot)$: *Terminal cost,*
  $u_\theta(\cdot, \cdot)$: *Initial parameterized policy,*
  $N$: *Number of trajectories sampled per update,*
  $\tau$: *Time horizon,*
  $\nu$: *Variance of Brownian noise,*
  $R$: *Control cost matrix,*
  $\mu$: *Learning rate,*
  $dt$: *Time discretization step*
**while** *not converged* **do**
  $\triangleright$ *Generate trajectories with current policy $u_\theta$*
  $\lambda \leftarrow R\nu$ ;
  $n \leftarrow 0$ ;
  **while** $n < N$ **do**
    $\triangleright$ *Initialize initial trajectory state*
    $(r_{n,0}, v_{n,0}, t) \leftarrow (r_0, \mathbf{0}, 0)$;
    **while** $t < (\tau/dt)$ **do**
      $\triangleright$ *Sample Brownian noise and action*
      $\varepsilon_{n,t} \sim \mathcal{N}(0, \nu)$;
      $u_{n,t} \leftarrow u_\theta(r_{n,t}, t)$;
      $\triangleright$ *Update positions and velocity*
      $r_{n,t+1} \leftarrow r_{n,t} + v_{n,t} \cdot dt$;
      $v_{n,t+1} \leftarrow v_{n,t} - \left(\nabla_{r_{n,t}} U(r_{n,t}) + u_{n,t} + \varepsilon_{n,t}\right) \cdot dt$;
      $t \leftarrow t + 1$;
    **end**
    $\triangleright$ *Determine trajectory cost and gradient*
    $C_n \leftarrow \frac{1}{\lambda}(\varphi(r_{n,\tau}) + \sum_{i=0}^{\tau} u_{n,i}^T R u_{n,i} + u_{n,i}^T R \varepsilon_{n,i})$;
    $\Delta\theta_n \leftarrow \exp(-C_n) + \sum_{i=0}^{\tau} \frac{\partial u_{n,i}}{\partial\theta}$;
    $n \leftarrow n + 1$ ;
  **end**
  $\triangleright$ *Determine gradient normalization and perform policy update*
  $\eta \leftarrow \sum_{i=0}^{N} \exp(-C_i)$;
  $\theta \leftarrow \theta + \frac{\mu}{\eta} \sum_{i=0}^{N} \Delta\theta_i$;
**end**

---

## A.2  SAMPLING

---

**Algorithm 2:** Sampling using parameterized control $u_\theta$

---

**Input:** $r_0$: *Initial molecular positions,*
  $U(\cdot)$: *Potential Energy function,*
  $u_\theta(\cdot, \cdot)$: *Trained parameterized policy,*
  $\tau$: *Time horizon,*
  $dt$: *Time discretization step*
$\triangleright$ *Initialize initial trajectory state*
$(r_t, v_t, t) \leftarrow (r_0, \mathbf{0}, 0)$;
**while** $t < (\tau/dt)$ **do**
  $\triangleright$ *Determine action*
  $u_t \leftarrow u_\theta(r_t, t)$;
  $\triangleright$ *Update positions and velocity*
  $r_{t+1} \leftarrow r_t + v_t \cdot dt$;
  $v_{t+1} \leftarrow v_t - (\nabla_{r_t} U(r_t) + u_t) \cdot dt$;
  $t \leftarrow t + 1$;
**end**

---

# B    STOCHASTIC OPTIMAL CONTROL

In this section we expand on Section 2.1. Specifically, we expand on the derivation of the Stochastic Optimal Control (SOC) objective in terms of a KL-divergence (appendix B.1) and the derivation of the iterative gradient descent procedure (appendix B.2). Note that the derivations presented here are a rephrasing of those given in (Kappen and Ruiz, 2016) using notation similar to the remainder of the paper. One difference to prior work can however be found in the relation between the distribution over uncontrolled and controlled dynamics.

Let us start by restating the objective of Path Integral Stochastic Optimal Control. Given a control $\boldsymbol{u}$ and the Brownian motion $\boldsymbol{\varepsilon}_t$ with variance $\nu$, Equation (4) defines a trajectory $\boldsymbol{x}(\tau) = (\boldsymbol{x}_0, \ldots, \boldsymbol{x}_\tau) \in \mathbb{R}^{r \times d}$. We can define the cost for said trajectory as

$$C(\boldsymbol{x}(\tau), \boldsymbol{u}, \boldsymbol{\varepsilon_t}) = \frac{1}{\lambda}\left(\varphi(\boldsymbol{x}_\tau) + \sum_{t=0}^{\tau} \frac{1}{2}\boldsymbol{u}(\boldsymbol{x}_t, t)^T \boldsymbol{R}\boldsymbol{u}(\boldsymbol{x}_t, t) + \boldsymbol{u}(\boldsymbol{x}_t, t)^T \boldsymbol{R}\boldsymbol{\varepsilon}_t\right) \tag{14}$$

where $\varphi$ denotes the terminal cost, $\lambda$ is a constant and $\boldsymbol{R}$ defines a weighted control cost.

This is a restatement of Equation (5), included here to make future reference easier. We note a number of important observations.

1. Following eq. (4), we observe that the control $\boldsymbol{u}$ acts *linearly* on the dynamics of the system.
2. The cost of the control itself is *quadratic*, weighted by the matrix $\boldsymbol{R}$.
3. Under expectation the final term vanished; $\mathbb{E}_{\boldsymbol{\varepsilon}}[\boldsymbol{u}(\boldsymbol{x}_t, t)^T \boldsymbol{R}\boldsymbol{\varepsilon}_t] = 0$

The first two observations are what classify the current control problem in the family of *Path Integral Stochastic Optimal Control* (Kappen, 2005) and are a requirement to be able to derive the explicit expression for the optimal control policy given in eq. (10). The third observation, while unusual in the context of SOC, is needed to rewrite the SOC objective in terms of the KL-divergence as we will see next. Additionally, if we restate the SOC-objective

$$\boldsymbol{u}^* = \arg\min_{\boldsymbol{u}} \mathbb{E}_{\tau, \boldsymbol{\varepsilon_t}}[C(\boldsymbol{x}(\tau), \boldsymbol{u}, \boldsymbol{\varepsilon_t})] \tag{15}$$

we observe that observation 3 shows that the additional cost does not change the optimal control $\boldsymbol{u}^*$.

Lastly, we note that the family of Path Integral Stochastic Optimal Control problems assumes that $\lambda = \boldsymbol{R}\nu$. This assumption is needed both for rewriting the SOC objective as a KL-divergence and to find an explicit expression for the solution.

## B.1    SOC OBJECTIVE AS A KL-DIVERGENCE

As noted in the main body of the paper, an adjustment needs to be made to Equation (7) due to the incorporation of the second-order dynamics. For this purpose we restate eq. (13) as

$$\boldsymbol{x}_{t+1} = \begin{pmatrix} \boldsymbol{r}_{t+1} \\ \boldsymbol{v}_{t+1} \end{pmatrix} = \begin{pmatrix} \boldsymbol{r}_t \\ \boldsymbol{v}_t \end{pmatrix} + \underbrace{\begin{pmatrix} \boldsymbol{v}_t \\ -\nabla_{\boldsymbol{r}_t}\boldsymbol{U}(\boldsymbol{r}_t) \end{pmatrix}}_{\boldsymbol{f}(\boldsymbol{x}_t, t)} \mathrm{d}t + \underbrace{\begin{pmatrix} \boldsymbol{0}_{3n} \\ \mathbb{I}_{3n} \end{pmatrix}}_{\boldsymbol{G}(\boldsymbol{x}_t, t)} \cdot \Big(\boldsymbol{u}(\boldsymbol{x}_t, t)\,\mathrm{d}t + \mathrm{d}\boldsymbol{\varepsilon}_t\Big), \tag{16}$$

with $t \in [0, \tau]$. We observe here that given $\boldsymbol{r}_t$ and $\boldsymbol{v}_t$, $\boldsymbol{r}_{t+1}$ and $\boldsymbol{v}_{t+1}$ are conditionally independent. As such we can derive a factorized probability distribution $\pi_{\boldsymbol{u}}(\boldsymbol{x}(\tau))$ over trajectories $\boldsymbol{x}(\tau)$ as

$$\pi_{\boldsymbol{u}}(\boldsymbol{x}(\tau)) = \pi_{\boldsymbol{u}}^{\boldsymbol{r}}(\boldsymbol{x}(\tau)) \cdot \pi_{\boldsymbol{u}}^{\boldsymbol{v}}(\boldsymbol{x}(\tau)) \tag{17}$$

with

$$\pi_{\boldsymbol{u}}^{\boldsymbol{r}}(\boldsymbol{x}(\tau)) = \prod_{t=0}^{\tau} \mathbb{1}_{[\boldsymbol{r}_{t+1}=\boldsymbol{r}_t+\boldsymbol{v}_t]}(\boldsymbol{r}_{t+1}) \tag{18}$$

$$\pi_{\boldsymbol{u}}^{\boldsymbol{v}}(\boldsymbol{x}(\tau)) = \prod_{t=0}^{\tau} \mathcal{N}(\boldsymbol{v}_{t+1}|\boldsymbol{\mu}_t, \Sigma_t). \tag{19}$$

Here, the normal distribution describing the transition probabilities for the velocity component is similar to eq. (4) with $\boldsymbol{\mu}_s = \boldsymbol{v}_s + \boldsymbol{f_v}(\boldsymbol{x}_s, s)\,\mathrm{d}t + \boldsymbol{G_v}(\boldsymbol{x}_s, s)(u(\boldsymbol{x}_s, s)\,\mathrm{d}t)$ and $\Sigma_s = \boldsymbol{G_v}(\boldsymbol{x}_s, s)^T \nu \boldsymbol{G_v}(\boldsymbol{x}_s, s)$. With $\boldsymbol{f}_v$ and $\boldsymbol{G}_v$ we denote the components of $\boldsymbol{f}$ and $\boldsymbol{G}$ acting on the velocity, respectively $-\nabla_{\boldsymbol{r}_t}\boldsymbol{U}(\boldsymbol{r}_t)$ and $\mathbb{I}_{3n}$.

Similarly, Equation (3) defines a probability distribution $\pi_0\big(\boldsymbol{x}(\tau)\big)$, where now $\boldsymbol{u} = 0$:

$$\pi_0\big(\boldsymbol{x}(\tau)\big) = \pi_0^{\boldsymbol{r}}\big(\boldsymbol{x}(\tau)\big) \cdot \pi_0^{\boldsymbol{v}}\big(\boldsymbol{x}(\tau)\big) \tag{20}$$

with $\pi_{\boldsymbol{u}}^{\boldsymbol{r}}\big(\boldsymbol{x}(\tau)\big) = \pi_0^{\boldsymbol{r}}\big(\boldsymbol{x}(\tau)\big)$ and

$$\pi_0^{\boldsymbol{v}}\big(\boldsymbol{x}(\tau)\big) = \prod_{t=0}^{\tau} \mathcal{N}(\boldsymbol{x}_{t+1}|\hat{\boldsymbol{\mu}}_t, \hat{\Sigma}_t) \tag{21}$$

with $\hat{\boldsymbol{\mu}}_s = \boldsymbol{v}_s + \boldsymbol{f_v}(\boldsymbol{x}_s, s)\,\mathrm{d}t$ and $\hat{\Sigma}_s = \Sigma_s$.

Because we are only interested in the relation between $\pi_{\boldsymbol{u}}\big(\boldsymbol{x}(\tau)\big)$ and $\pi_0\big(\boldsymbol{x}(\tau)\big)$, the same analysis as in (Kappen and Ruiz, 2016) applies with $\pi_{\boldsymbol{u}}^{\boldsymbol{r}}\big(\boldsymbol{x}(\tau)\big)$ and $\pi_0^{\boldsymbol{r}}\big(\boldsymbol{x}(\tau)\big)$ cancelling out. Following Girsanov (Cameron and Martin, 1944), we get:

$$\pi_{\boldsymbol{u}}(\boldsymbol{x}(\tau)) = \pi_0(\boldsymbol{x}(\tau)) \exp\Big( \sum_{t=0}^{\tau} -\frac{1}{2}\frac{\boldsymbol{u}_t^T \boldsymbol{G}_t^T \boldsymbol{G}_t \boldsymbol{u}_t}{\Sigma_t} + \frac{\boldsymbol{G}_t \boldsymbol{u}_t(\boldsymbol{f}_t + \boldsymbol{v}_t - \boldsymbol{v}_{t-1})}{\Sigma_t} \Big)$$

$$= \pi_0(\boldsymbol{x}(\tau)) \exp\Big( \sum_{t=0}^{\tau} -\frac{1}{2}\frac{\boldsymbol{u}_t^T \boldsymbol{G}_t^T \boldsymbol{G}_t \boldsymbol{u}_t}{\Sigma_t} + \frac{\boldsymbol{G}_t \boldsymbol{u}_t(\boldsymbol{G}_t(\boldsymbol{u}_t + \boldsymbol{\varepsilon}_t))}{\Sigma_t} \Big)$$

$$= \pi_0(\boldsymbol{x}(\tau)) \exp\Big( \sum_{t=0}^{\tau} \frac{1}{2}\frac{\boldsymbol{u}_t^T \boldsymbol{G}_t^T \boldsymbol{G}_t \boldsymbol{u}_t}{\Sigma_t} + \frac{\boldsymbol{u}_t^T \boldsymbol{G}_t^T \boldsymbol{G}_t \boldsymbol{\varepsilon}_t}{\Sigma_t} \Big)$$

$$= \pi_0(\boldsymbol{x}(\tau)) \exp\Big( \sum_{t=0}^{\tau} \frac{1}{2}\boldsymbol{u}_t^T \boldsymbol{G}_t^T \Sigma_t^{-1} \boldsymbol{G}_t \boldsymbol{u}_t + \boldsymbol{u}_t^T \boldsymbol{G}_t^T \Sigma_t^{-1} \boldsymbol{G}_t \boldsymbol{\varepsilon}_t \Big)$$

$$= \pi_0(\boldsymbol{x}(\tau)) \exp\Big( \sum_{t=0}^{\tau} \frac{1}{2}\boldsymbol{u}_t^T \nu^{-1} \boldsymbol{u}_t + \boldsymbol{u}_t^T \nu^{-1} \boldsymbol{\varepsilon}_t \Big)$$

$$= \pi_0(\boldsymbol{x}(\tau)) \exp\Big( \frac{1}{\lambda}\sum_{t=0}^{\tau} \frac{1}{2}\boldsymbol{u}_t^T \boldsymbol{R} \boldsymbol{u}_t + \boldsymbol{u}_t^T \boldsymbol{R}\boldsymbol{\varepsilon}_t \Big), \tag{22}$$

where we use the assumption $\lambda = \boldsymbol{R}\nu$ in the last step. We use shorthand notation to simplify $\boldsymbol{u}_t = u(\boldsymbol{x}_t, t)$, $\boldsymbol{G}_t = \boldsymbol{G_v}(\boldsymbol{x}_t, t)$, and $\boldsymbol{f}_t = \boldsymbol{f_v}(\boldsymbol{x}_t, t)$. From here we can obtain the relation in Equation (8).

Now, as again show in (Kappen and Ruiz, 2016), we can use this relation to rewrite the cost in eq. (14) as

$$C(\boldsymbol{x}(\tau), \boldsymbol{u}, \boldsymbol{\varepsilon_t}) = \frac{1}{\lambda}\varphi(\boldsymbol{x}_\tau) + \log \frac{\pi_{\boldsymbol{u}}\big(\boldsymbol{x}(\tau)\big)}{\pi_0\big(\boldsymbol{x}(\tau)\big)}, \tag{23}$$

and thus, the distribution over trajectories under optimal control $\boldsymbol{u}^*$ can now be defined as

$$\pi_{\boldsymbol{u}^*} = \arg\min_{\pi_{\boldsymbol{u}}} \mathbb{E}_{\boldsymbol{x}(\tau)\sim\pi_{\boldsymbol{u}}}[C(\boldsymbol{x}(\tau), \boldsymbol{u}, \boldsymbol{\varepsilon_t})]$$

$$= \arg\min_{\pi_{\boldsymbol{u}}} \mathbb{E}_{\boldsymbol{x}(\tau)\sim\pi_{\boldsymbol{u}}}\Big[\frac{1}{\lambda}\varphi(\boldsymbol{x}_\tau) + \log \frac{\pi_{\boldsymbol{u}}\big(\boldsymbol{x}(\tau)\big)}{\pi_0\big(\boldsymbol{x}(\tau)\big)}\Big]$$

$$= \arg\min_{\pi_{\boldsymbol{u}}} \mathbb{E}_{\boldsymbol{x}(\tau)\sim\pi_{\boldsymbol{u}}}\Big[\frac{1}{\lambda}\varphi(\boldsymbol{x}_\tau)\Big] + \mathbb{E}_{\boldsymbol{x}(\tau)\sim\pi_{\boldsymbol{u}}}\Big[\log \frac{\pi_{\boldsymbol{u}}\big(\boldsymbol{x}(\tau)\big)}{\pi_0\big(\boldsymbol{x}(\tau)\big)}\Big]$$

$$= \arg\min_{\pi_{\boldsymbol{u}}} \mathbb{E}_{\boldsymbol{x}(\tau)\sim\pi_{\boldsymbol{u}}}\Big[\frac{1}{\lambda}\varphi(\boldsymbol{x}_\tau)\Big] + \mathbb{D}_{\mathsf{KL}}\Big(\pi_{\boldsymbol{u}}\big(\boldsymbol{x}(\tau)\big)\|\pi_0\big(\boldsymbol{x}(\tau)\big)\Big) \tag{24}$$

This objective is an approximation of the Schrodinger Bridge formulation in Equation (2) where the constraints on the marginal distributions are replaced by a regularization term in the form of the terminal cost. When the expected terminal cost dominates the KL-divergence the found distribution should be similar.

## B.2 ITERATIVE GRADIENT DESCENT

As mentioned earlier, the specific control problem we are considering here (linear acting control and weighted quadratic control cost) is known as *Path Integral Stochastic Optimal Control*. Work on this control problem has established that under the additional assumption that $\lambda = \boldsymbol{R}\nu$ there exists an explicit solution describing the optimal control $\boldsymbol{u}^*$. While there are a number of different papers establishing this result (Kappen, 2005; Theodorou et al., 2010; Kappen, 2007), we note that (Kappen and Ruiz, 2016) is most in line with our work. As such, we refer the interested reader to this work to find the proof for the following statement that defines the distribution over optimal trajectories as a reweighing of the distributions over trajectories under no control:

$$\pi_{\boldsymbol{u}^*} = \frac{1}{\eta}\pi_0\big(\boldsymbol{x}(\tau)\big)\exp(-\frac{1}{\lambda}\varphi(\boldsymbol{x}_\tau)), \tag{25}$$

where $\eta = \mathbb{E}_{\boldsymbol{x}(\tau)\sim\pi_0}[\exp(-\frac{1}{\lambda}\varphi(\boldsymbol{x}_\tau)]$ is the normalization constant. Given the previously established relation (eq. (8)) between $\pi_0$ and $\pi_{\boldsymbol{u}}$, we can equivalently express the optimal control $\boldsymbol{u}^*$ in terms of any control $\boldsymbol{u}$ using importance sampling

$$
\begin{aligned}
\pi_{\boldsymbol{u}^*} &= \frac{1}{\eta}\frac{\pi_0\big(\boldsymbol{x}(\tau)\big)}{\pi_{\boldsymbol{u}}\big(\boldsymbol{x}(\tau)\big)}\pi_{\boldsymbol{u}}\big(\boldsymbol{x}(\tau)\big)\exp(-\frac{1}{\lambda}\varphi(\boldsymbol{x}_\tau)) \\
&= \frac{1}{\eta}\frac{1}{\exp\left(\frac{1}{\lambda}\sum_{t=0}^{\tau}\frac{1}{2}\boldsymbol{u}_t^T\boldsymbol{R}\boldsymbol{u}_t + \boldsymbol{u}_t^T\boldsymbol{R}\boldsymbol{\varepsilon}_t\right)}\pi_{\boldsymbol{u}}\big(\boldsymbol{x}(\tau)\big)\exp(-\frac{1}{\lambda}\varphi(\boldsymbol{x}_\tau)) \\
&= \frac{1}{\eta}\pi_{\boldsymbol{u}}\big(\boldsymbol{x}(\tau)\big)\exp(-\frac{1}{\lambda}\varphi(\boldsymbol{x}_\tau) - \frac{1}{\lambda}\sum_{t=0}^{\tau}\frac{1}{2}\boldsymbol{u}_t^T\boldsymbol{R}\boldsymbol{u}_t - \boldsymbol{u}_t^T\boldsymbol{R}\boldsymbol{\varepsilon}_t) \\
&= \frac{1}{\eta}\pi_{\boldsymbol{u}}\big(\boldsymbol{x}(\tau)\big)\exp(-C(\boldsymbol{x}(\tau), \boldsymbol{u}, \boldsymbol{\varepsilon}_t))
\end{aligned}
\tag{26}
$$

With an explicit expression for the optimal control policy given, the PICE method aims to find a parameterized distribution $\pi_{\boldsymbol{u}_{\theta^*}}$ that is close to the optimal control in terms of KL-divergence

$$\pi_{\boldsymbol{u}_\theta^*} = \underset{\pi_{\boldsymbol{u}_\theta}}{\arg\min}\,\mathbb{D}_{\mathsf{KL}}\Big(\pi_{\boldsymbol{u}^*}\big(\boldsymbol{x}(\tau)\big)\|\pi_{\boldsymbol{u}_\theta}\big(\boldsymbol{x}(\tau)\big)\Big). \tag{27}$$

Using the explicit expression for the optimal control, the KL-divergence is given as follows:

$$
\begin{aligned}
\mathbb{D}_{\mathsf{KL}}&\Big(\pi_{\boldsymbol{u}^*}\big(\boldsymbol{x}(\tau)\big)\|\pi_{\boldsymbol{u}_\theta}\big(\boldsymbol{x}(\tau)\big)\Big) \\
&\propto -\mathbb{E}_{\pi_{\boldsymbol{u}^*}}\big[\log\pi_{\boldsymbol{u}_\theta}\big] \\
&= -\mathbb{E}_{\pi_{\boldsymbol{u}^*}}\Big[\log\pi_0(\boldsymbol{x}(\tau))\exp\Big(\sum_{t=0}^{\tau}-\frac{1}{2}\frac{\boldsymbol{u}_\theta(t)^T\boldsymbol{G}_t^T\boldsymbol{G}_t\boldsymbol{u}_\theta(t)}{\Sigma_t} + \frac{\boldsymbol{G}_t\boldsymbol{u}_\theta(t)(f_t + \boldsymbol{x}_t - \boldsymbol{x}_{t-1})}{\Sigma_t}\Big)\Big] \\
&\propto -\mathbb{E}_{\pi_{\boldsymbol{u}^*}}\Big[\sum_{t=0}^{\tau}-\frac{1}{2}\frac{\boldsymbol{u}_\theta(t)^T\boldsymbol{G}_t^T\boldsymbol{G}_t\boldsymbol{u}_\theta(t)}{\Sigma_t} + \frac{\boldsymbol{G}_t\boldsymbol{u}_\theta(t)(\boldsymbol{G}_t(\boldsymbol{u}^*(t) + \boldsymbol{\varepsilon}_t))}{\Sigma_t}\Big] \\
&= \mathbb{E}_{\pi_{\boldsymbol{u}^*}}\Big[\frac{1}{\lambda}\sum_{t=0}^{\tau}\frac{1}{2}\boldsymbol{u}_\theta(t)^T\boldsymbol{R}\boldsymbol{u}_\theta(t) - \boldsymbol{u}_\theta(t)^T\boldsymbol{R}\boldsymbol{u}^*(t) - \boldsymbol{u}_\theta(t)^T\boldsymbol{R}\boldsymbol{\varepsilon}_t\Big] \\
&= \frac{1}{\eta}\mathbb{E}_{\pi_{\boldsymbol{u}}}\Big[e^{-C(\boldsymbol{x}(\tau),\boldsymbol{u},\boldsymbol{\varepsilon}_t)}\frac{1}{\lambda}\sum_{t=0}^{\tau}\frac{1}{2}\boldsymbol{u}_\theta(t)^T\boldsymbol{R}\boldsymbol{u}_\theta(t) - \boldsymbol{u}_\theta(t)^T\boldsymbol{R}\boldsymbol{u}(t) - \boldsymbol{u}_\theta(t)^T\boldsymbol{R}\boldsymbol{\varepsilon}_t\Big] \quad (28)
\end{aligned}
$$

We use shorthand notation to simplify $\boldsymbol{u}(t) = \boldsymbol{u}(\boldsymbol{x}_t, t)$, $\boldsymbol{G}_t = \boldsymbol{G}(\boldsymbol{x}_t, t)$, and $f_t = f(\boldsymbol{x}_t, t)$. Line 1 we discard the constant term $\mathbb{E}_{\pi_{\boldsymbol{u}^*}}\big[\log\pi_{\boldsymbol{u}^*}\big]$. Line 2 we make use of the established relation between $\pi_{\boldsymbol{u}}$ and $\pi_0$ for any control $\boldsymbol{u}$. Line 3 we discard the constant term $\mathbb{E}_{\pi_{\boldsymbol{u}^*}}\big[\log\pi_0\big]$ and note that the expectation is over trajectories sampled from $\pi_{\boldsymbol{u}^*}$. Line 4 we rewrite using the assumption that $\lambda = \boldsymbol{R}\nu$. Line 5 we use Equation (26) to rewrite the distribution over an arbitrary control $\boldsymbol{u}$ using.

We can minimize this explicit expression using Gradient Descent, to do this, we derive the gradient of the KL-divergence

$$
\frac{\partial \mathbb{D}_{\mathsf{KL}} \left( \pi_{\boldsymbol{u}^*} \left( \boldsymbol{x}(\tau) \right) \| \pi_{\boldsymbol{u}_\theta} \left( \boldsymbol{x}(\tau) \right) \right)}{\partial \theta}
$$
$$
= \frac{1}{\eta} \mathbb{E}_{\pi_{\boldsymbol{u}}} \left[ e^{-C(\boldsymbol{x}(\tau), \boldsymbol{u}, \boldsymbol{\varepsilon}_t)} \frac{1}{\lambda} \sum_{t=0}^{\tau} (\boldsymbol{R} \boldsymbol{u}_\theta(t) - \boldsymbol{R} \boldsymbol{u}(t) - \boldsymbol{R} \boldsymbol{\varepsilon}_t) \frac{\partial \boldsymbol{u}_\theta(t)}{\partial \theta} \right]. \tag{29}
$$

Finally, we note that the expectation is over trajectories of any distribution $\pi_{\boldsymbol{u}}$, and as such, this distribution can also be chosen to be equal to the parameterized distribution $\pi_{\boldsymbol{u}} = \pi_{\boldsymbol{u}_\theta}$. This gives us the final gradient

$$
\frac{\partial \mathbb{D}_{\mathsf{KL}} \left( \pi_{\boldsymbol{u}^*} \left( \boldsymbol{x}(\tau) \right) \| \pi_{\boldsymbol{u}_\theta} \left( \boldsymbol{x}(\tau) \right) \right)}{\partial \theta}
$$
$$
= \frac{1}{\eta} \mathbb{E}_{\pi_{\boldsymbol{u}_\theta}} \left[ e^{-C(\boldsymbol{x}(\tau), \boldsymbol{u}, \boldsymbol{\varepsilon}_t)} \frac{1}{\lambda} \sum_{t=0}^{\tau} (\boldsymbol{R} \boldsymbol{u}_\theta(t) - \boldsymbol{R} \boldsymbol{u}_\theta(t) - \boldsymbol{R} \boldsymbol{\varepsilon}_t) \frac{\partial \boldsymbol{u}_\theta(t)}{\partial \theta} \right] \tag{30}
$$
$$
= -\frac{1}{\eta} \mathbb{E}_{\pi_{\boldsymbol{u}_\theta}} \left[ e^{-C(\boldsymbol{x}(\tau), \boldsymbol{u}_\theta, \boldsymbol{\varepsilon}_t)} \frac{1}{\lambda} \sum_{t=0}^{\tau} \boldsymbol{R} \boldsymbol{\varepsilon}_t \frac{\partial \boldsymbol{u}_\theta(t)}{\partial \theta} \right]. \tag{31}
$$

## C  EXTENSION EXPERIMENTAL SECTION

### C.1  OPENMM

**General setup:**  We use the Velocity Verlet with Velocity Randomization (VVVR) integrator (Sivak et al., 2013) within OpenMM at a temperature of $300\,\mathrm{K}$ with a collision rate of $1.0\,\mathrm{ps}^{-1}$.

**Alanine Dipeptide:**  We use the amber 99sb-ildn force field (Lindorff-Larsen et al., 2010) without any solvent, a time-step of $1.0\,\mathrm{fs}$ for the VVVR integrator and a cutoff of $1\,\mathrm{nm}$ for the Particle Mesh Ewald (PME) method (Essmann et al., 1995).

**Polyproline Helix:**  We initialize OpenMM with the amber protein.ff14SBonlysc forcefield and gbn2 as the implicit solvent forcefield. The VVVR integrator had a timestep of $2.0\,\mathrm{fs}$ and a cutoff of $5\,\mathrm{nm}$ for PME. The proposed method was ran for a total of $10.000\,\mathrm{fs}$ (resulting in 5,000 policy steps).

**Chignolin:**  To sample transition paths between the folded and unfolded state of the Chignolin protein, we initialize OpenMM using the same forcefield and VVVR integrator as for Polyproline with the exception that we sample a new force from our policy network every $1.0\,\mathrm{fs}$. We do this 5000 times for each rollout for a total time horizon of $5000\,\mathrm{fs}$.

### C.2  ALANINE DIPEPTIDE

#### C.2.1  DISCUSSION BASELINES AND EVALUATION METRICS

**Metrics**  Three different metrics are used for the comparison covering multiple desiderata for the sampled transition trajectories. For each metric we report the score over 1000 trajectories with the exception of the *Molecular Dynamics without fixed timescale* baseline which is only ran until 10 trajectories are successfully generated.

*Expected Pairwise Distance (EPD)* The EPD measures the similarity between the final conformation in the trajectory and the target conformation taking into account the full 3D geometry of the molecule. Note that the expected pairwise distance for uncontrolled MD with the target as the starting conformation has a EPD of $2.25 \times 10^{-3}$. All trajectories with an EPD of less than this can thus be considered to transition the molecule within one standard deviation of the target distribution.

*Target Hit Percentage (THP):* The second metric under which we evaluate the proposed Transition Path Sampler measures the similarity of the final and target conformation in terms of the collective

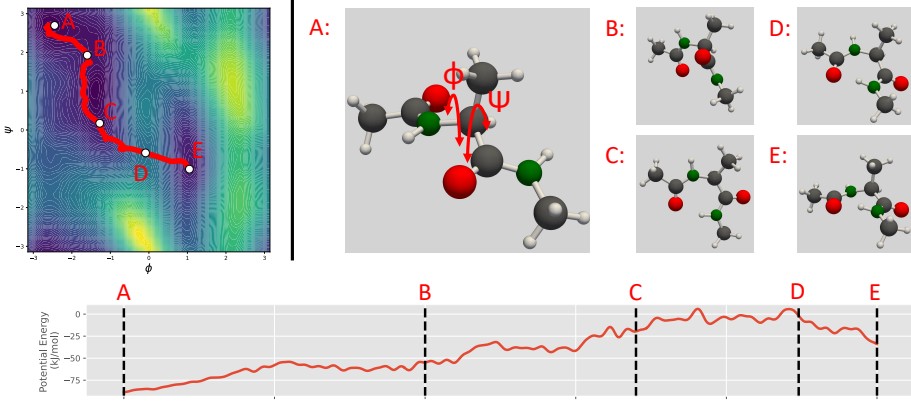

Figure 4: Visualization of a trajectory sampled with the proposed force prediction method. *Left:* The sampled trajectory projected on the free energy landscape of Alanine Dipeptide as a function of two CVs *Right:* Conformations along the sampled trajectory: A) starting conformation showing the CV dihedral angles, B-D) intermediate conformations with D being the highest energy point on the trajectory, and E) final conformation, which closely aligns with the target conformation. *Bottom:* Potential energy during transition. Letters represent the same configurations in the transition.

variables. The THP measures the percentage of generated trajectories/paths that reach the target state. As such, higher hit percentages are preferred. We determine a trajectory to have hit the target in CV space when $\phi$ and $\psi$ are both within 0.75 of the target.

*Energy Transition Point (ETP):* The final metric looks at the potential energy of the transition point—the conformation in the trajectory with the highest potential energy. This directly evaluates the capability of the method to find the transition path that crosses the boundary at the lowest saddle point.

**Baselines** We compare the proposed Transition Path Sampling method with extended Molecular Dynamics simulation using different time-scales and temperature points. As discussed earlier, there are currently no other methods available for Transition Path Sampling using the full 3D geometry of the molecules.

*Molecular Dynamics with fixed timescale:* This set of baselines is limited to the same timescale as the proposed Transition Path Sampler, 500 femtoseconds, but uses varying temperatures. With higher temperatures we should have a higher probability of crossing the barrier and hitting the target configuration.

*Molecule Dynamics without fixed timescales:* In contrast to the other set of baselines, the MD simulation for this set is not limited to 500 femtoseconds, but is instead ran until the target conformation is reached. We consider a trajectory to have reached its target if the following two conditions have been met: 1) the current conformation classifies as having hit the target under the conditions of the metric described above and 2) the current conformation is within one standard deviation of the target distributions mean.

By running the MD simulations until the target is reached we aim to gain intuition into the speed-up that it achieved by the fixed timescale of the proposed Transition Path Sampler.

### C.2.2 ADDITIONAL RESULTS: VISUALIZATION FORCE PREDICTION

We observe that the force predicting policy has learned a different trajectory then the energy predicting model presented in the main body of the paper. While different, both of the trajectories pass the high energy barrier in a locally low point. Previous work on finding transition path has also observed that multiple viable paths can be found for Alanine Dipeptide (Hooft et al., 2021).

