# OpenReview forum: "PIPS: Path Integral Stochastic Optimal Control for Path Sampling in Molecular Dynamics"
_ICLR.cc/2023/Conference — Submitted to ICLR 2023_

### Official Review · Reviewer_RDYQ · 2022-10-24

**Confidence:** 4
**Correctness:** 3
**Technical Novelty And Significance:** 1
**Empirical Novelty And Significance:** 2
**Recommendation:** 3

**Clarity, Quality, Novelty And Reproducibility:**

Clarity:
- paper is overall understandable, with many points under-defined or un-defined however:
- Eq (2) lacks context and it seem like pi* could just equal pi
- Eq (5) mentions the matrix R, which is equite mysterious to me. In the code however it is equal to the Identity matrix I ... which such complications to end up using I ?  This looks like obfuscation.
- what is lambda, how is interpreted ?
- terminal cost is not defined (can be guessed, but only at page 5..)
- The way in which the policy is learned is not explained at all. MLPs are mentioned in page 6. What is the task, input, output, etc, is not explained at all.
- I do not understand how the learning accounts for what will happen in the future, in uncertain (noise) conditions. This is very mysterious to me, how a policy which only depends on t, can learn and work (it does seem to work !)

Quality
- overall, the scientific quality is ok, but then most things are recalls of prior works, not original work.
- the important fact that terminal cost dominates eq (9) is not obvious at all.

Novelty:
- The paper very much lacks novelty. I detailed this at length above in Weaknesses.
- As remark 3 says:
    > This connection between the Schrödinger Bridge Problem and stochastic optimal control has been previously established (Chen et al., 2016; Pavon et al., 2021). However, through the formulations in Equations (2) and (9), we also establish the equivalence between sampling transition paths, Schrödinger bridge problem, and stochastic optimal control.


Reproducibility:
- the code is available anonymously, seems well packaged.

**Strength And Weaknesses:**

Strengths:
- the notations are overall clear, the content is understandable and concisely reported.
- the experiments are abundantly commented (in terms of commenting the results)

Weaknesses:
- Machine Learning, or Learning Representation, seems quite lacking from the paper. Neural networks are mentioned quickly, only on page 5.
- mostly, the lack of novelty, in several points. I detail here:
- I recall that Eq (2) is the Schrödinger Bridge Problem (SBP). Then Eq (9), which is claimed to establish equivalence between sampling transition paths, Schrödinger bridge problem, and stochastic optimal control, however, has the term of terminal cost inside it, making it different from (2). Paper says that:
     > Therefore, when the terminal cost dominates the KL-divergence term above, it enforces the target boundary constraints of the problem.

     But in practice, in most problems of finding good transition paths, the initial and final configurations' energies are quite similar, and it's not obvious that the "terminal cost" dominates (I realized afterwards I probably inferred incorrectly what is the terminal cost.. it should be defined early and intuitively in the paper. If its expression depends on how we parameterize the configuration... I do not understand well how it can relate with weights of paths

- algorithmic novelty: here again, PIPS does not seem to significantly differ from PICE (Hilbert Johan Kappen and Hans Christian Ruiz. Adaptive importance sampling for control and inference. Journal of Statistical Physics, 162(5):1244–1266, 2016). Actually the code submitted contains the file PICE.py, which performs Algorithm A.1 (page 13). Indeed, the authors are very honest about it:
     > we note that (Kappen and Ruiz, 2016) is most in line with our work
- on the theory side, the paper seems to be strongly inspired (and it is not hidden at all, on the contrary, which is very much honorable) by works from Kappen:  Kappen, 2005; 2007; Kappen and Ruiz, 2016. All equations/theory/results stated up to end of sec. 2 are attributed to other papers, and mostly these 2 papers.
- On the interpretation side, similarly, Das 2021 and Rose 2021 (including Garrahan) are cited, but not exploited. By very quickly looking at these references, it seems like a good deal of the intuitive physics (that is I think only partially explained in the paper), are explained over there. For instance, papers says:
   > Remark 2. The last term in the cost function in eq. (5) relating the Brownian motion and the control is unusual and devoid of a clear intuition.
    I think intuition comes from considering, where could this external force we apply to the molecule, come from, in the original (no external force) setup ? I think that the answer is the noise: it can "conspire" to correlate to be equal or close to the control policy, .. if one is very lucky (hence the rare transitions are rare !)
- as I said in the summary, the main (only?) novelty is the use of second order Hamiltonian dynamics, i.e. sec 3, which is in 1 page long, in particular Eq (13). However, this seems to me like a rather classic trick to assemble position and velocity, in order to make a Dynamical System become Markov-like, under time discretization. However, I recognize that if it's the needed step to apply new techniques (SOC, SBP, PICE) to transition path sampling, well then, good job. (this could become a strength if I am convinced this is very much new and insightful compared to previous works).
- even sec. 3 is inspired from refs:
     > We, thus, take inspiration from recent computational advances for solving SBP (Vargas et al., 2021a; De Bortoli et al., 2021) to develop our solution in §3 to solve the problem of sampling transition paths that can efficiently cross the high free energy barriers

    but I did not have time to check these references.


**Summary Of The Paper:**

The paper proposes a method for finding the most likely transition between two states in high-dimensional configuration spaces (for chemical compounds, typically). It does so on building on recent advances in the Schrödinger bridge problem (SBP) and stochastic optimal control (SOC).

It claims to show equivalence between sampling transition paths (the most likely one) and these two other equivalent problems (SBP and SOC).

The main novelty is the use of second order Hamiltonian dynamics, i.e. sec 3, in particular Eq (13) (which seems to me like a rather classic trick to assemble position and velocity to make a Dynamical System become Markov-like, under time discretization).

Another novelty, possibly (I am not so aware of the literature) is the very use of SOC / RL-style approaches to transition path sampling. But, this does not seem to be altogether new: people are discussing the use of RL for path sampling, which is very close to the proposed approach (apart from finite vs infinite time horizon).

The proposed approach is tested on 3 classic transition-paths small molecules, with some success (but no comparison with other similar approaches, as Reinforcement Learning).

**Summary Of The Review:**

Overall, this paper is rather sound and rather clear, but lacks novelty, by its own account, and additionally does not focus on ML at all.

Hence, I lean to reject, quite strongly, until further arguments are given about the originality of the approach and its merit for the community.

---

> ### Author Response · Authors · 2022-11-17
> **Response to reviewer  RDYQ**
>
> We would like to thank the reviewer for taking the time to review our paper. Before addressing some points from the review directly, we would like to clarify the two main issues the review states with our work; missing ML / LR focus and the lack of novelty.
>
> ### General Comments
> **Regarding ML / LR focus:**
>
> The reviewer states:
>
> > Machine Learning, or Learning Representation, seems quite lacking from the paper. Neural networks are mentioned quickly, only on page 5.
>
> We disagree with this statement from the reviewer. While our work focuses on a subject area outside mainstream machine learning, great care was taken to align the (computational) chemistry problem with terminology familiar to the machine learning audience. It is for this reason that the paper steps away from traditional discussion of minimal energy paths, and instead takes a KL-Divergence minimization approach (a problem fundamental in machine learning).
>
> Furthermore, the main objective of the paper is to find a probability distribution $\pi_{u_{\theta}}$ (parameterized by a neural network) that best approximates $\pi$ (the unbiased dynamics) under some constraints (the marginal distributions at time step 0 and $\tau$) given simulated data. We believe this to be a very fundamental process in machine learning.
>
> **Regarding the lack of novelty:**
>
> We submitted our paper under the category of "Machine Learning for Sciences (eg biology, physics, health sciences, social sciences, climate/sustainability)" and can as such be considered an application paper. Please note that under the ICLR Call-For-Papers, this is explicitly listed as a relevant topic for the conference.
>
> Additionally, while it is true that our paper primarily focuses on the application of an existing approach to a novel subject area, we believe that the review does not sufficiently take into account that the application in itself is not trivial. As outlined in the paper, there are a number of important considerations that had to be taken into account that made it possible for our Machine Learning based approach to be applied to the problem of sampling transitions paths:
>
> 1. For use in downstream applications, the generated transitions have to be Minimal Energy Trajectories, meaning that the total sum of the applied control should be minimized. This is a hard requirement for our problem often not satisfied in standard RL approaches (rewards are not related to the magnitude of the action), or other optimal control approaches (for example, [1] does not minimise the amount of control directly).
>
> 2. For practical purposes, the control needs to be able to interface with traditional MD simulation software such as OpenMM. This requires redefining the control to linearly act as a bias potential through the velocity component of second-order dynamics used in MD simulation. While this seems straightforward, this is to our knowledge the first work that restructures the control problem as finding a bias potential to be able to interface with MD simulations.
>
> While we understand that these considerations look straightforward in hindsight, we can assure the reviewer that they are not. This is exemplified by the fact that earlier work on SBP and optimal control has not been able to apply work to the field of sampling paths in Molecular transitions, despite the importance of this problem. (In private communication with others working on the Schrodinger Bridge Problem the interface with MD simulation has been cited as a major hurdle for applying these approaches to the problem of sampling molecular transition paths)
>
> It is also for this reason that there is no comparison to prior work using RL (or other approaches) for sampling molecular transition paths. We do not wish to claim that our Path Integral-based approach is the best machine learning solution for sampling transition paths between molecular confirmations, but at the time of writing, we are the only one. We hope that our work can serve as a strong starting point for further exploration of this problem using machine learning approaches.
>
> [1] Theodorou, E., Buchli, J., & Schaal, S. (2010). A generalized path integral control approach to reinforcement learning. The Journal of Machine Learning Research, 11, 3137-3181.

---

> > ### Author Response · Authors · 2022-11-17
> > **Response to reviewer RDYQ - 2**
> >
> > ### Specific comments
> >
> > > Q1: But in practice, in  most problems of finding good transition paths, the initial and final  configurations' energies are quite similar, and it's not obvious that  the "terminal cost" dominates (I realized afterwards I probably inferred incorrectly what is the terminal cost.. it should be defined early and  intuitively in the paper. If its expression depends on how we  parameterize the configuration... I do not understand well how it can  relate with weights of paths
> >
> > A1: The section referred to by the reviewer is a general introduction to the PICE method and therefore does not directly mention the loss function used. The statement regarding the domination of the terminal cost is independent of the actual implementation. We deliberately tried to separate the discussion of the method itself and the implementation based on the problem to not cause confusion, but we can add a foreshadowing note near equation 5 if the review believes this will improve readability.
> >
> > > Q2: PIPS does not seem to significantly differ from PICE
> >
> > A2: PIPS can be considered a variant of PICE with specific conditions met to be applicable to the sampling of molecular transitions. It is important to note, however, that the considerations made (eg. an equivariant terminal cost) allow for scaling PICE beyond previously attempted problem dimensionality.
> >
> > > Q3: All equations/theory/results stated up to end of sec. 2
> >
> > A3: While this is primarily true, we would like to refer the reviewer to the paragraph before Remark 4 which clarifies that the theory is not directly applicable without addressing the issue with the singular covariance matrix.
> >
> > > Q4: similarly, Das 2021 and Rose 2021 (including Garrahan) are cited, but not exploited.
> >
> > A4: Both Das and Rose consider the use of Reinforcement Learning for sampling rare transitions between molecular confirmations. However, neither of the papers discusses the problem in terms of describing a bias potential for the entire system and instead focuses on a lower dimensional sub-problem of the problem. (This is similar to considering the lower dimensional Collective Variable based approach) This is another strong indicator that the application of PICE to this domain is in itself not trivial.
> >
> > >Q5: where could this  external force we apply to the molecule, come from, in the original (no  external force) setup ? I think that the answer is the noise: it can  "conspire" to correlate to be equal or close to the control policy,
> >
> > A5: This is correct, without the external force setup, the transitions occur due to the entropic terms in the molecular free energy. The policy we learn therefore aims to mimic the components of the noise that are responsible for driving the transition. This is what the learning achieves by generating random noisy trajectories and updating the policy based on the most "effective" noisy trajectories.
> > The statement regarding the relationship between the Brownian motion and the control in remark 2 is however related to the standard Path Integral stochastic optimal control definition which does not contain such a term.
> >
> > >Q6: However, this seems  to me like a rather classic trick to assemble position and velocity, in  order to make a Dynamical System become Markov-like, under time  discretization. However, I recognize that if it's the needed step to  apply new techniques (SOC, SBP, PICE) to transition path sampling, well  then, good job.
> >
> > A6: This is partly correct, the combination of position and velocity is an important step to be able to apply new techniques to the transition path sampling. However, this glosses over the reconstruction of the control as a bias potential acting on the system. While the Markov-like interpretation is a requirement, it is this linear control interpretation as a bias potential that allows for effectively integrating with MD simulations.
> >
> > Concluding, yes, this reconstruction is an important contribution of the work and one of the main reasons why prior work did not consider this problem before.
> >
> > >Q7: even sec. 3 is inspired from refs:
> >
> > A7: Note that the main purpose of these references is to relate the SBP to sampling transition paths to justify the use of Path Integral control, these references do not discuss how to construct the control as a bias potential.
> >
> > >Q8: Eq (2) lacks context and it seem like pi* could just equal pi
> >
> > A8: Below equation 2 we defined the space over which $\pi^*$ is defined. Considering the constraints put on this space, $\pi^* = \pi$ is only possible when $\pi$ itself has the required marginals, in which case the problem would be non-existent.

---

> > > ### Author Response · Authors · 2022-11-17
> > > **Response to reviewer RDYQ - 3**
> > >
> > > >Q9: Eq (5) mentions the matrix R, which is equite mysterious to me. In the code however it is equal to the Identity matrix I ... which such complications to end up using I ? This looks like obfuscation.
> > >
> > > A9: Matrix R is part of the formal definition of Path Integral control. As explained, it can be interpreted as the weight matrix for the quadratic control. While the method can deal with arbitrary positive-definite matrices $R$, for our purpose we need the weights to be equal for all molecules and thus use $I$. This is not a matter of obfuscation, but rather an attempt to keep the method construction general.
> > >
> > > >Q10: what is lambda, how is interpreted ?
> > >
> > > A10: Similar to above, lambda is a general term from Path Integral control theory. It is important for rewriting the SOC objective in terms of KL-divergences as discussed above equation 8.
> > >
> > > >Q11: The way in which the  policy is learned is not explained at all. MLPs are mentioned in page 6. What is the task, input, output, etc, is not explained at all.
> > >
> > > A11: This is discussed in the "Optimal Control Policy" part of section 2.1 where we explicitly state the update rule for minimizing the KL-divergence.
> > >
> > > >Q12: I do not understand  how the learning accounts for what will happen in the future, in  uncertain (noise) conditions. This is very mysterious to me, how a  policy which only depends on t, can learn and work (it does seem to work !)
> > >
> > > A12: The policy does not only depend on the time, but rather on the state at time step $t$. This can be seen from the definition of the second-order dynamics in Eq. 13 where $u$ takes both $x_t$ and $t$ as input.
> > >
> > > >Q13: the important fact that terminal cost dominates eq (9) is not obvious at all.
> > >
> > > A13: The remark concerning eq. 9 does not state that the terminal cost dominates the control cost, but rather that if the terminal cost dominates the problem reduces to the SBP.
> > >
> > > >Q14: As remark 3 says:
> > >
> > > A14: Please note the second part of this remark. While the connection between SBP and optimal control is from prior work, relating this to the problem of transition path sampling is not.

---

### Official Review · Reviewer_XQgJ · 2022-10-25

**Confidence:** 4
**Correctness:** 3
**Technical Novelty And Significance:** 2
**Empirical Novelty And Significance:** 3
**Recommendation:** 5

**Clarity, Quality, Novelty And Reproducibility:**

The paper is clear, and the application of the existing theory to molecular systems is novel, but the underlying methods have been used before.

**Strength And Weaknesses:**


The loss fucntion that is optimized is a combination of two:

Distance between desired and actual descriptors at the end of the trajectory: This part of the loss is unarguable. The trajectories minimizing this part are obviously transition paths.

A penalty term for the magnitude of the control: This is less justified.
There is a very strong assumption underneath that the weight of the penalty is directly proportional to the noise. In a theoretical investigation of control this is fine. In a case when we one does not seek physically ideal bias and control strength e.g. when first part of the loss is much more important that the second one, this is also fine. But in case of transition paths and potentially free energy estimates - as proposed in the paper outline - such simplification does not stand and is likely too simple to produce realiable estimates of the potential of the means force.

The gradient update is the weighted average of the trajectories, reinforcing those that had low loss and forgetting the one with high loss. This will always produce a path connecting A and B. But in what sense is this a transition path? Are tehre guarantees that the underlying physics are respected? What would be the result if one optimized just MSE of the (actual - desired) descriptors in the final point?

Appendix A1 There is possibly a typo in \Delta \Theta_n -> .... line. Is the exponential weight with the control gradient or added to?

The authors claim they cannot compare the method to any other. I think the method potentially could be compared to differentiable simulation of protein folding, using codes like TorchMD or JAXMD.

The simulations construct what are essentially transition paths, but are not called so. If this method produces better TS paths then the second part of the loss helps in some way.

It might be easier then to explain the difference between figure 1 and 4 if they were plotted together. Illustrative molecules can be omitted for on of the results as there is not much insight gained from them being just printed.


**Summary Of The Paper:**

This paper proposes a control algorithm to explore transition paths between reactant and product basins in molecular dynamics simulations. The work is strictly based on the theory of Knappen and Ruiz (2016), and implements it for molecular systems.


**Summary Of The Review:**

The approach seems to generate a reasonable path in energy landscape. The proposed approach can be extraordinarily aggressive and may not work well with temperature or to produce potential of mean force. It may be useful for people trying to find transition paths but will likely require downstream refinement to be practically useful.

---

> ### Author Response · Authors · 2022-11-17
> **Response to reviewer XQgJ**
>
> We thank the reviewer for taking the time to review the paper. Unfortunately, we believe that the reviewer has misunderstood some key components of our contribution. We hope that the discussion below can clarify some of the misconceptions.
>
> > Q1: But in case of transition paths and potentially free energy estimates - as proposed in the paper outline - such simplification does not stand and is likely too simple to produce realiable estimates of the potential of the means force.
>
> A1: We disagree with the reviewer on this for the following reason. In the PIPS framework, the control acts linearly on the velocities of the molecular dynamics, and can as such be considered as a bias force derived as the gradient of the bias potential. Crucially, this means that by minimizing the amount of control applied (through the quadratic control cost) we are directly minimizing the bias force and thus the bias potential. While the terminal cost dominates the control cost and places the emphasis on the optimization of generating acceptable transition paths (adhering to the boundary conditions), the control cost will still force minimal energy paths. The two controls (terminal and cost) are not in conflict and can both be satisfied at the same time.
>
> Regarding the potential of the mean force; we note that our work applies a bias force to every single atom and as such, the description of a potential of mean force is hard to define. An in-depth discussion of the relation between our bias potential and the potential of mean force was consciously left out of the paper to focus on the technical machine learning component of the contribution.
>
> >Q2: But in what sense is this a transition path?
>
> A2: The terminal cost forces them to be transition paths between conformation A and B. Additionally, the control cost forces these transition paths to be minimal energy paths.
>
> >Q3: Are tehre guarantees that the underlying physics are respected?
>
> A3: Yes, the paths are generated using MD simulation software and can thus be considered to respect the underlying physics used in these simulations. Additionally, the control cost forces the transition steps to be of high probability (low bias force).
>
> >Q4: Is the exponential weight with the control gradient or added to?
>
> A4: Thank you for bringing this to our attention, the loss should be used as a weight. We will address this in the paper update.
>
> >Q5: I think the method potentially could be compared to differentiable simulation of protein folding, using codes like TorchMD or JAXMD. The simulations construct what are essentially transition paths, but are not called so. If this method produces better TS paths then the second part of the loss helps in some way.
>
> A5: To clarify, packages such as TorchMD and JAXMD are neural network approximations of molecular simulation software such as OpenMM (which we used for our work). These packages themselves do therefore not allow for sampling transition paths between molecular conformations with high free energy barriers between them.
> The experiment described by the reviewer, to sample transitions between two conformations using TorchMD and JAXMD, would therefore be the same as we present in Table 1. Without our bias potential, these packages would only sample transitions when the temperature is increased. As Table 1 shows, these transitions would not be minimal (higher ETP).
>
> >Q6: Illustrative molecules can be omitted for on of the results as there is not much insight gained from them being just printed.
>
> A6: We included the illustrative molecules to highlight that besides the dihedral angles being outside of equilibrium, the remainder of the molecules is still physically correct. Given the reviewer's concern regarding the physical reliability of our method, we believe this to be an important part of the visualization.

---

### Official Review · Reviewer_5HjD · 2022-10-31

**Confidence:** 3
**Correctness:** 4
**Technical Novelty And Significance:** 2
**Empirical Novelty And Significance:** 3
**Recommendation:** 5

**Clarity, Quality, Novelty And Reproducibility:**

I found the manuscript quite easy to read and the experiments clearly described. In terms of novelty, this seems like a slight modification to an existing method for application in an interesting domain. As such I was slightly surprised by the relatively small amount of empirical (quantitative) evaluation.

This paper could also use some more specificity for additional reproducibility. For example it is stated that “The width of the layers of the policy network is dependent on the number of atoms in the molecule under consideration.” What were these widths and how were they determined? Code (which is not available for review as far as I’m aware) could potentially help here.

**Strength And Weaknesses:**

Strengths:

- Interesting application of optimal control
- Neat parameterization in terms of second-order dynamics that allows easy integration with OpenMM.
- Well explained experiments which are understandable by non MD experts

Weaknesses:

- Limited theoretical novelty as the theory is established in prior work.
- Limited quantitative experiments on a single (simple) molecule. As a non-expert in MD I can’t evaluate how often good CVs are available and how much they help in this path sampling problem. Do the authors think an experiment with MD using knowledge of the CVs be useful? It would help me understand how this baseline method works. Presumably when the CVs are less accurate (such as in chignolin?) CV-based optimizations would not work as well as PIPS? Could this be confirmed experimentally?
- Another concern here is overfitting. How are we sure that the parameters are not simply overfit to each molecule making it difficult to apply to new molecule simulations? It would be helpful to state how hyperparameters were tuned and on what data.

Comments:

I don’t think I understand remark 3. Is this a claim that equations (2) and (9) are the first to establish the equivalence between sampling transition paths and the SB problem which hasn’t been previously established? I don’t understand how these equations are meant to establish this and would be very surprised if this is the first time this equivalence has been established.

The notation in the “Physics inspired policy network” is a bit confusing to me with bold $u_\theta$ and regular $u_\theta$ having very different (and unspecified) dimensions. Instead of defining F(r_t) could we say instead that bold $u_\theta$ is either directly parameterized (force prediction) or parameterized as $\nabla_{r_t} E_\theta$ (energy prediction)?

I understand the current evaluation is on the quality of the paths. I would also be curious as to the quality in terms of the optimization, i.e. what is the value of $E_{\tau, \epsilon_t} C(x(\tau), u, \epsilon_t)$ for the force vs. energy models?

$u_\theta$ with some invariances is mentioned, but the end network is an MLP. Was a more sophisticated network tried? Perhaps it is not helpful for these single-molecule studies.

Question: How does Temperature come in to play? Its slightly odd to me that MD without a fixed timescale is evaluated on a different temperature than PIPS. Could PIPS be evaluated on the same temperatures or could the authors explain why 300K is “better”?

Minor remarks:

Remark 3? is this a novelty claim?

misplaced comma last paragraph of page 5.

Courtsey —> Curtesy Remark 4

It would be useful to mention Table 1 is on Alanine Dipeptide in the caption. Also it slightly strange that it comes before figure 1, but is referenced afterwards.

**Summary Of The Paper:**

This work considers the problem of sampling transition paths between two metastable states of a molecular system. This problem is difficult as the energy barrier between the states may be large, making it computationally expensive for traditional MD simulation. This work first relates this problem to literature on optimal control and Schrodinger bridges then provides a parameterization in terms of second-order dynamics to learn likely transition paths. This method is demonstrated on three systems using two slightly different parameterizations, either directly modelling the control force, or indirectly modelling the force as $\nabla E$ and directly parameterizing energy.

**Summary Of The Review:**

I found this paper interesting and timely with the increased interest in diffusion-based models. For me I would like to see more experimental validation and rigor in terms of quantitative comparisons and experimental setup.

---

> ### Author Response · Authors · 2022-11-17
> **Response to reviewer 5HjD**
>
> We would like to thank the reviewer for their time reviewing our paper. We especially appreciated the reviewer's comment regarding the understandability of the paper for non-MD experts. While writing the manuscript significant effort was taken to make the paper accessible for both the MD and ML community.
>
> Before discussing individual comments in detail, we would like to clarify some concerns regarding the novelty of our work.
>
> **General comments regarding novelty**:
>
> First of all, we would like to highlight that we submitted our paper under the of "Machine Learning for Sciences (eg biology, physics, health sciences, social sciences, climate/sustainability)" and can as such be considered an application paper. Please note that under the ICLR Call-For-Papers, this is explicitly listed as a relevant topic for the conference.
>
> The reviewer is correct in assuming that Remark 3 is a statement to claim the novelty of building the relationship between sampling molecular transition paths and the relevant machine learning literature. While previous work has connected the Schrodinger bridge problem and optimal control, it has not extended the relationship to the application area of interest here. Formally establishing this connection is an important contribution of the work as it allows for applying solutions from these fields to the problem at hand.
>
> Furthermore, in addition to establishing this connection, an important contribution is the integration of the ML approaches with the MD simulations. To answer the reviewer's question; parameterization as a bias potential is not only a nice interpretation, it is crucial for applying these solutions.

---

> > ### Author Response · Authors · 2022-11-17
> > **Response to reviewer 5HjD - 2**
> >
> > **Specific comments:**
> >
> > >Q1: As a non-expert in MD I can’t evaluate how often good CVs are available and how much they help in this path sampling problem.
> >
> > A1: When CVs are available, they can speed up the sampling of transition paths as they greatly reduce the dimensionality of the problem. However, good CVs are hardly ever present for molecules of practical interest. It is therefore crucial to develop methods that don't depend on them.
> >
> > >Q2: Do the authors think an experiment with MD using knowledge of the CVs be useful?
> >
> > A2: When including CVs in the problem, it fundamentally changes the underlying assumptions. Specifically, when introducing the CV as the action space in which the sampling process operates the direct connection with the minimal energy paths is lost. In this setting, the control does not naturally relate to a bias potential anymore.
> >
> > >Q3: Presumably when the CVs are less accurate (such as in chignolin?) CV-based optimizations would not work as well as PIPS?
> >
> > A3: Yes, this is correct. Consider the Alanine Dipeptide example. If we remove one of the angles, the transition path would be forced to cross over higher energy barriers.
> >
> > >Q4: Could this be confirmed experimentally?
> >
> > A4: This could be confirmed, but this is outside of the scope of our work. The use of CV and their influence on sampled trajectories is a separate field of study in computational chemistry.
> >
> > >Q5: How are we sure that  the parameters are not simply overfit to each molecule making it  difficult to apply to new molecule simulations? It would be helpful to state how hyperparameters were tuned and on what data.
> >
> > A5: To clarify, a new policy is trained for each molecule following the training procedure in section 3. For each molecule, this does require some hyperparameter tuning such as learning rate and model size. However, we found that these hyperparameters only differed slightly between the molecules studied. A simple hand search was sufficient.
> >
> > In addition to that, there are a number of chemically inspired hyperparameters such as the transition duration $\tau$ which has to be set based on experience. This is however more accessible than the setting of CVs.
> >
> > >Q6: Instead of defining F(r_t) could we say instead that bold  is either directly parameterized (force prediction) or parameterized as  (energy prediction)?
> >
> > A6: We would like to thank the reviewer for this suggestion. We will rewrite this section to remove the introduction of $F$.
> >
> > >Q7: I would also be curious as to the quality in terms of the optimization, i.e. what is the value of  for the force vs. energy models?
> >
> > A7: Unfortunately, we don’t have these exact values ready at the moment. However, these values are closely related to the presented EDP and ETP scores. A lower EDP represents a lower terminal loss, and a lower ETP represents a lower control cost. Based on the results presented in Table 1, we can thus conclude that the energy model would have a lower trajectory cost.
> >
> > >Q8: Was a more sophisticated network tried? Perhaps it is not helpful for these single-molecule studies.
> >
> > A8: This is part of a larger follow-up study.
> >
> > >Q9: How does Temperature come in to play?
> >
> > A9: This is an important question, thank you for asking. The temperature of the simulation is directly related to the noise level of the sampled trajectories. As such, with higher temperatures, there is a higher probability of sampling transition paths between the initial and target states. However, under high temperatures, these are noisy transitions and might thus not relate to minimal energy transitions.
> >
> > For finding minimal energy paths, we thus want to sample them with a temperature as low as possible (300K is a normal temperature to run these simulations at). Unfortunately, using low temperatures it is very unlikely to sample the rare transitions, and thus we need to increase the temperature when running MD without our bias potential. This is exemplified by the THP metric being almost 0 for all MD simulations with a fixed timescale, and the successful trajectories (both with a fixed and without a fixed timescale) having a very high ETP.
> >
> > >Q10: Minor remarks:
> >
> > A10: Thank you for these comments, we will address them in the update.
> >
> > >Q11: This paper could also use some more specificity for additional reproducibility. [...] Code (which is not available for review as far as I’m aware) could potentially help here.
> >
> > A11: The code is linked in the paper in section 4 ( https://github.com/pips4anonymous/pips-anonymous). If there is anything left unclear with the given code, we are happy to address further questions.
> >
> > >Q12: For me I would like to see more experimental validation and rigor in terms of quantitative comparisons and experimental setup
> >
> > A12: We hope the clarifications regarding the nature of the temperature used in the simulation and linking the code in our rebuttal have resolved some of the concerns around the experimental rigour.

---

### Decision · Program_Chairs · 2023-01-20

**Decision:**

Reject

**Justification For Why Not Higher Score:**

As noted by several reviewers, while there is clear motivation for the overall work, and this work has very good presentation, the method is based on established theoretical work so thus is a relatively straightforward idea, and on the application side there are still limitations as the reviewers noted that would need to be addressed. The authors could consider alternative forum for presenting this work if they would like to focus more on the application side of this topic.


**Justification For Why Not Lower Score:**

N/A

**Metareview: Summary, Strengths And Weaknesses:**

The current paper propose a method for sampling transition paths across two meta-stable states when typically done using expensive, resource-intensive MD simulation method for a large macro-molecular system. The author demonstrated their approach using a set of examples. There is several merits of this work, including the well-motivated question that is being addressed here, the clear presentation of their methods and good demonstration and implementation. The weaknesses are mainly focused on the potential issues such as the overall novelty of this method, and if considering the application side, how broadly and useful the method would be across different settings.